# Exploring Geometry of Blind Spots in Vision Models

**Sriram Balasubramanian**[*]  **Gaurang Sriramanan**[*]  **Vinu Sankar Sadasivan**
sriramb@cs.umd.edu     gaurangs@cs.umd.edu     vinu@cs.umd.edu

**Soheil Feizi**
sfeizi@cs.umd.edu

Department of Computer Science
University of Maryland, College Park

## Abstract

Despite the remarkable success of deep neural networks in a myriad of settings, several works have demonstrated their overwhelming sensitivity to near-imperceptible perturbations, known as adversarial attacks. On the other hand, prior works have also observed that deep networks can be under-sensitive, wherein large-magnitude perturbations in input space do not induce appreciable changes to network activations. In this work, we study in detail the phenomenon of under-sensitivity in vision models such as CNNs and Transformers, and present techniques to study the geometry and extent of "equi-confidence" level sets of such networks. We propose a Level Set Traversal algorithm that iteratively explores regions of high confidence with respect to the input space using orthogonal components of the local gradients. Given a source image, we use this algorithm to identify inputs that lie in the same equi-confidence level set as the source image despite being perceptually similar to arbitrary images from other classes. We further observe that the source image is linearly connected by a high-confidence path to these inputs, uncovering a star-like structure for level sets of deep networks. Furthermore, we attempt to identify and estimate the extent of these connected higher-dimensional regions over which the model maintains a high degree of confidence. The code for this project is publicly available at this URL.

## 1 Introduction

Deep neural networks have demonstrated remarkable success in various diverse domains including computer vision and natural language processing, surpassing the performance of classical algorithms by a significant margin. However, though they achieve state of the art results on many computer vision tasks like image classification, sometimes even exceeding human-level performance [He et al., 2015, 2016], the overall visual processing conducted by such models can deviate significantly from that effectively observed in the human visual system. Perhaps most iconic and representative of these differences lies in the overwhelming susceptibility of neural networks to near-imperceptible changes to their inputs — commonly called adversarial attacks [Szegedy et al., 2014] — an extraordinary failure mode that highlights the *over-sensitivity* of such models. Indeed, an active area of research over the past few years has been focused towards analyzing adversarial perturbations under different threat models [Goodfellow et al., 2015, Tramèr et al., 2018, Wong et al., 2019, Laidlaw et al., 2021, Croce and Hein, 2022] and in addressing adversarial vulnerabilities using robust training methodologies [Madry et al., 2018, Zhang et al., 2019b, Singla and Feizi, 2020, Wu et al., 2020, Laidlaw et al., 2021, Levine and Feizi, 2021].

---

[*]Equal Contribution, author names ordered alphabetically

37th Conference on Neural Information Processing Systems (NeurIPS 2023).

On the other hand, it has been shown that such models may also be *under-sensitive*, wherein input images that are unmistakably disparate to a human oracle induce near identical network activations or predictions [Jacobsen et al., 2018a]. To tractably analyze this phenomenon, Jacobsen et al. [2018a] utilize a special class of neural networks that are bijective functions, called fully Invertible RevNets [Jacobsen et al., 2018b, Kingma and Dhariwal, 2018], to craft large-magnitude semantic perturbations in input space that are designed to leave its corresponding network representations unchanged. Furthermore, Tramèr et al. [2020] show that robust training with $\ell_p$ bounded adversaries can be a source of excessive model-invariance in itself, due to the poor approximation of the true imperceptible threat model of human oracles by $\ell_p$ norm-bounded balls in RGB-pixel space [Laidlaw et al., 2021]. Indeed, the authors 'break' a provably robust defense on MNIST [Zhang et al., 2019a] with a certified accuracy of $87\%$, by crafting perturbations within the certified $\ell_\infty$ radius of $0.4$, that however cause model agreement with human oracles to diminish to $60\%$.

However, these methods either rely upon special invertible network architectures or the selection of the nearest training image of another class as a target followed by a sequence of complex alignment and spectral clustering techniques, so as to semantically alter the input in a conspicuous manner that induces a change in the human assigned oracle label, while leaving the model prediction unchanged. This leads us to our research question: Is it possible to analyze the phenomenon of *under-sensitivity* of general vision models in a systematic manner on natural image datasets, and characterize the geometry and extent of "blind spots" of such networks? Indeed, we empirically demonstrate the veracity of this claim — in this work, we present a novel Level Set Traversal algorithm to explore the "equi-confidence" level sets of popular vision models. Given an arbitrary source and target image pair, our proposed algorithm successfully finds inputs that lie in the same level set as the source image, despite being near-identical perceptually to the target image. Furthermore, the proposed algorithm identifies a connected path between the original source image and the "blind spot" input so generated, wherein high prediction confidence with respect to the source class is maintained throughout the path. In summary, we make the following contributions in this work:

- We present a novel Level Set Traversal algorithm that iteratively uses orthogonal components of the local gradient to identify the "blind spots" of common vision models such as CNNs and ViTs on CIFAR-10 and ImageNet.
- We thereby show that there exist piecewise-linear connected paths in input space between images that a human oracle would deem to be extremely disparate, though vision models retain a near-uniform level of confidence on the same path.
- Furthermore, we show that the linear interpolant path between these images also remarkably lies within the same level set; as we observe the consistent presence of this phenomenon across arbitrary source-target image pairs, this unveils a star-like set substructure within these equi-confidence level sets.
- We demonstrate that adversarially robust models tend to be *under-sensitive* over subsets of the input domain that lie well beyond its original threat model, and display level-sets of high-confidence that extend over a significant fraction of the triangular convex hull of a given source image and arbitrary pair of target images.

## 2 Preliminaries

**Notation:** In this paper, we primarily consider the setting of classification with access to a labelled dataset. Let $\boldsymbol{x} \in \mathcal{X}$ denote $d$-dimensional input images, and let their corresponding labels be denoted as $y \in \{1, \ldots, N\}$. Let $f : \mathcal{X} \to [0, 1]^N$ denote the classification model considered, where $f(\boldsymbol{x}) = \left(f^1(\boldsymbol{x}), \ldots, f^N(\boldsymbol{x})\right)$ represents the softmax predictions over the $N$-classes. Further, let $C_f(\boldsymbol{x})$ be the argmax over the $N$-dimensional softmax output, representing the class predicted by the model for input $\boldsymbol{x}$. For a given data sample $(\boldsymbol{x}, y)$, let the cross-entropy loss achieved by the model be denoted as $CE(f(\boldsymbol{x}), y)$. Given a prediction confidence value $p \in [0, 1]$, and a class $j \in \{1, \ldots, N\}$ we define the Level Set $L_f(p, j)$, and Superlevel Set $L_f^+(p, j)$ for the function $f$ as follows:

$$L_f(p, j) = \{\boldsymbol{x} \in \mathcal{X} : f^j(x) = p\} \ , \ L_f^+(p, j) = \{\boldsymbol{x} \in \mathcal{X} : f^j(x) \geq p\}$$

Given a pair of inputs $\boldsymbol{x}_1$ and $\boldsymbol{x}_2$, we define the linear interpolant path between them as $P(\lambda; \boldsymbol{x}_1, \boldsymbol{x}_2) = \lambda \cdot \boldsymbol{x}_1 + (1 - \lambda) \cdot \boldsymbol{x}_2$, for $\lambda \in [0, 1]$. A given set $S$ is thus said to be convex if $P(\lambda; \boldsymbol{x}_1, \boldsymbol{x}_2) \in S$, $\forall \boldsymbol{x}_1, \boldsymbol{x}_2 \in S$ and $\lambda \in [0, 1]$. Further, a set $S$ is said to be *star-like* if there exists some $\mathbf{s}_0 \in S$ such that $P(\lambda; \mathbf{s}_0, \boldsymbol{x}) \in S$, $\forall \boldsymbol{x} \in S$ and $\lambda \in [0, 1]$.

## 2.1 Conjugate Nature of Adversarial and Confidence Preserving Perturbations

Given a classification model $f$ and a correctly classified benign input $(\boldsymbol{x}, y)$, an adversarial image is a specially crafted image $\widetilde{\boldsymbol{x}} = \boldsymbol{x} + \boldsymbol{\varepsilon}$ such that both $\boldsymbol{x}$ and $\widetilde{\boldsymbol{x}}$ appear near-identical to a human oracle, but induces the network to misclassify, i.e. $C_f(\boldsymbol{x} + \boldsymbol{\varepsilon}) \neq C_f(\boldsymbol{x}) = y$. To enforce imperceptibility in a computationally tractable manner, several adversarial threat models have been proposed, with the $\ell_2$ and $\ell_\infty$ norm constraint models being the most popular. Amongst the earliest adversarial attacks specific to the latter threat model was the Fast Gradient Sign Method (FGSM) attack, proposed by Goodfellow et al. [2015], wherein the adversarial perturbation is found by single-step direct ascent along the local gradient with pixel-wise clipping. A stronger multi-step variant of this attack called Iterated-FGSM (IFGSM) was later introduced by [Kurakin et al., 2017], wherein iterated gradient ascent is performed alternately with projection operations onto the constraint set. A popular variant, called the Projected Gradient Descent (PGD) attack was introduced by Madry et al. [2018] which incorporates a initial random perturbation to the clean image, which was observed to help mitigate gradient masking effects Kurakin et al. [2016]. A large class of adversarial attacks [Croce and Hein, 2020, Gowal et al., 2019, Carlini et al., 2019, Sriramanan et al., 2020] thus utilize perturbations parallel to the gradient, with appropriate projection operations to ensure constraint satisfaction.

In contrast, perturbations that leave the network prediction confidence unchanged, are locally orthogonal to the gradient direction. Indeed, for any differentiable function $g : \mathbb{R}^d \to \mathbb{R}$, denote the level set as $L_g(c)$ for a given output $c \in \mathbb{R}$. Let $\gamma(t) : [0,1] \to L_g(c)$ be any differentiable curve within the level set. Then, $g(\gamma(t)) = c \; \forall t \in [0,1]$. Thus, $\frac{d}{dt}(g(\gamma(t))) = 0 = \langle \nabla g(\gamma(t)), \gamma'(t) \rangle$, implying that $\gamma'(t)$ is orthogonal to $\nabla g(\gamma(t)), \;\; \forall t \in [0,1]$. Since this is true for *any* curve $\gamma$ contained in the level set, we conclude that the gradient vector is always perpendicular to the level set. Furthermore, we can additionally show that the level set $L_g(c)$ is often a differentiable submanifold, with mild additional conditions on the gradient. Indeed, a level set $L_g(c)$ is said to be *regular* if $\nabla g(\boldsymbol{x}) \neq \boldsymbol{0} \;\; \forall \boldsymbol{x} \in L_g(c)$.

**Lemma 1.** *If $g : \mathbb{R}^d \to \mathbb{R}$ is a continuously differentiable function, then each of its regular level sets is a $(d-1)$ dimensional submanifold of $\mathbb{R}^d$.*

We present the proof of Lemma 1 in Section A of the Appendix. We thus observe that adversarial perturbations, largely parallel to the gradient, are locally orthogonal to confidence preserving directions which correspond to the $(d-1)$ dimensional tangent space of the level set. We take inspiration from this observation to develop a general framework that applies to a broad class of differentiable neural networks, as opposed to previous works [Jacobsen et al., 2018a] that require the network to be invertible to identify confidence preserving perturbations.

We also remark that adversarial attacks and level set preserving perturbations are complementary from another perspective as well, as noted by prior works: the former attempts to find inputs that change model predictions without modifying human oracle assignments, while the latter attempts to keep network predictions unchanged though human oracles would likely change their original label assignment. Thus, both classes of adversarial attacks and level set preserving perturbations induce misalignment between oracle and model predictions, and cast light onto independent means of evaluating the coherence of models towards human expectations.

## 3 Proposed Method: Level Set Traversal

We now describe our algorithm for traversing the level set, which we call the Level Set Traversal (LST) algorithm (Algorithm 1). We try to find a path from a source image to a target image such that all points on that path are classified by the model as the source class with high confidence. Given that these $(d-1)$ dimensional level set submanifolds can be potentially highly complex in their geometries, we use a discretized approximation using small yet finite step sizes to tractably explore these regions. Let the source image be $\boldsymbol{x}_s$ with true label $y_s$, and $\boldsymbol{g} = \nabla_{\boldsymbol{x}} CE(f(\boldsymbol{x}), y_s)$ be the gradient of the cross entropy loss of the model ($f$) prediction with respect to $\boldsymbol{x}$. The key idea is to get as close as possible to a target image $\boldsymbol{x}_t$ from some image $\boldsymbol{x}$ by computing the projection of $\Delta\boldsymbol{x} = \boldsymbol{x}_t - \boldsymbol{x}$ onto the orthogonal complement of $\boldsymbol{g}$, to obtain a new image $\boldsymbol{x}_{\text{new}}$ that leaves the model confidence unchanged. We can compute the projection of $\Delta\boldsymbol{x}$ on the orthogonal complement by subtracting the component of $\Delta\boldsymbol{x}$ parallel to $\boldsymbol{g}$, that is, $\boldsymbol{x}_{||\boldsymbol{g}} = \boldsymbol{g}\left(\frac{\Delta\boldsymbol{x}^\top \boldsymbol{g}}{\|\boldsymbol{g}\|^2}\right)$ (L6). Then, using a scale factor $\eta$, the vector update to $\boldsymbol{x}$ can be expressed as $\Delta\boldsymbol{x}_\perp = \eta(\Delta\boldsymbol{x} - \boldsymbol{x}_{||\boldsymbol{g}})$ (L7), and therefore $\boldsymbol{x}_{\text{new}} = \boldsymbol{x} + \Delta\boldsymbol{x}_\perp$. Starting from the source image $\boldsymbol{x}_s$, we repeatedly perform this iteration to get reasonably close to the target image

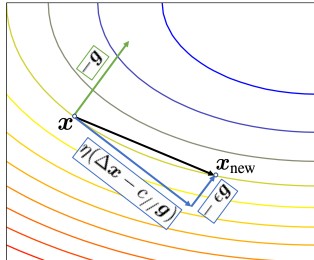

Figure 1: Pictorial representation of the LST algorithm. The contour lines represent level sets of the model confidence $f(\boldsymbol{x})$, with blue representing high confidence (low loss) and red representing low confidence (high loss). At each iteration, we obtain $\boldsymbol{x}_{\text{new}}$ by adding two vectors, $\boldsymbol{x}_\perp = \eta(\Delta\boldsymbol{x} - c_{//}\boldsymbol{g})$ (projection of $\Delta\boldsymbol{x}$ onto the orthogonal complement of $\boldsymbol{g}$) and $\boldsymbol{x}_{||}$ (a small perturbation to increase the confidence and remain within the level set).

---

**Algorithm 1** Level Set Traversal (LST)

---

1: **Input:** Source image $\boldsymbol{x}_s$ with label $y$, target image $\boldsymbol{x}_t$, model $f$, max iterations $m$, scale factor $\eta$, stepsize $\epsilon$, confidence threshold $\delta$
2: Initialize $\boldsymbol{x} = \boldsymbol{x}_s, \boldsymbol{x}_{||} = \boldsymbol{0}$
3: **for** $i = 1$ **to** $m$ **do**
4: $\quad \Delta\boldsymbol{x} = \boldsymbol{x}_t - \boldsymbol{x}$
5: $\quad \boldsymbol{g} = \nabla_{\boldsymbol{x}}CE(f(\boldsymbol{x}), y)$
6: $\quad c_{//} = (\boldsymbol{g} \cdot \Delta\boldsymbol{x})/||\boldsymbol{g}||^2$
7: $\quad \Delta\boldsymbol{x}_\perp = \eta(\Delta\boldsymbol{x} - c_{//}\boldsymbol{g})$
8: $\quad \boldsymbol{x}_{||} = \Pi_\infty(\boldsymbol{x}_{||} - \epsilon\boldsymbol{g}, -\epsilon, \epsilon)$
9: $\quad \boldsymbol{x}_{\text{new}} = \boldsymbol{x} + \Delta\boldsymbol{x}_\perp + \boldsymbol{x}_{||}$
10: $\quad$ **if** $f(\boldsymbol{x}_s)[j] - f(\boldsymbol{x}_{\text{new}})[j] > \delta$ **then**
11: $\quad\quad$ **return** $\boldsymbol{x}$
12: $\quad \boldsymbol{x} = \boldsymbol{x}_{\text{new}}$
13: **return** $\boldsymbol{x}$

---

| Step: 0 | Step: 10 | Step: 20 | Step: 40 | Step: 60 | Step: 80 | Step: 120 | Step: 160 | Step: 200 | Step: 300 | Step: 400 |
|---|---|---|---|---|---|---|---|---|---|---|
| 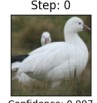 | 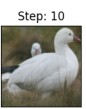 | 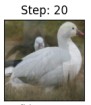 | 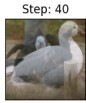 | 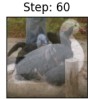 | 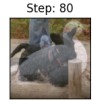 | 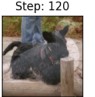 | 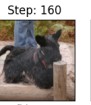 | 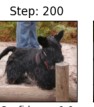 | 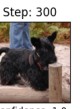 | 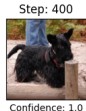 |
| Confidence: 0.997 | Confidence: 0.998 | Confidence: 0.999 | Confidence: 0.999 | Confidence: 1.0 | Confidence: 1.0 | Confidence: 1.0 | Confidence: 1.0 | Confidence: 1.0 | Confidence: 1.0 | Confidence: 1.0 |

Figure 2: Intermediate images over the path traversed by the LST algorithm using a source image of a 'goose' and a target image of a 'Scottish terrier' (a dog) for a normally trained ResNet-50 model. We observe that the model predicts the 'goose' class with very high confidence for all images over the path, though the target blindspot found by LST clearly appears as a 'dog' to human oracles.

while carefully ensuring that the confidence of the model prediction at any given point does not drop below a preset confidence threshold $\delta$ compared to the source image (L10).

While the above method works well with a relatively large $\eta$ if the curvature of $f$ is low, it risks a non-trivial drop in model confidence (or even a change in the model prediction) if the curvature of $f$ at $\boldsymbol{x}$ is high enough. Therefore, after each step, we add a small perturbation vector $\boldsymbol{x}_{||}$ in the direction of $-\boldsymbol{g}$ scaled by a factor of $\epsilon$. This step decreases the cross entropy loss, and thus increases the confidence so as to offset any confidence drops incurred due to the addition of $\Delta x_\perp$. Thus, we can maintain a higher model confidence over the path. In order to ensure that the norm of this perturbation is not arbitrarily large, we bound the $\ell_\infty$ norm by projecting the vector to an $\ell_\infty$ ball of radius $\epsilon$. Then, the step $\boldsymbol{x}_{||}$ can be computed as clamp$(\epsilon\boldsymbol{g}, -\epsilon, \epsilon)$, where the function clamp$(\boldsymbol{v}, a, b)$ is the element-wise application of the function $\min(\max(\cdot, a), b)$ on all elements of $\boldsymbol{v}$. Thus, we modify L9 so that now $\boldsymbol{x}_{\text{new}} = \boldsymbol{x} + \Delta\boldsymbol{x}_\perp - \boldsymbol{x}_{||}$. We present a pictorial schematic of Algorithm 1 in Fig 1. We further employ an exponential moving average of $\boldsymbol{x}_{||}$, in order to smoothen components that potentially undulate heavily during the iterative traversal. Thus, since the frequent changes in $\boldsymbol{x}_{||}$ are smoothened out, we find that the final output images are often linearly connected to the source image with high confidence over the linear interpolations (see Fig 6).

We now apply this algorithm to explore the level sets of standard and robust ResNet-50 models. In Fig 2, we present the path traversed by the LST algorithm, wherein the model predicts the source 'goose' class with very high confidence over the entire path, though the target blindspot found by LST clearly appears as a 'dog' to human oracles. To substantiate the efficacy of the LST algorithm, we randomly select five images from five arbitrary ImageNet classes ('goose', 'Scottish Terrier', 'meerkat', 'academic gown', 'cleaver'), and compute LST blindspots for all possible source-target image pairs. We show the final images output by the LST algorithm in Fig 3 as an image-grid. If we order the five selected images, the $i^{th}$ row and $j^{th}$ column of the image-grid is the LST output obtained using the $i^{th}$ image as target and $j^{th}$ image as the source. Thus, each column represents the source image being transformed iteratively into the other target images. The confidence of the model prediction for the source class (names on top of each column) is displayed just below each image. For both the normally trained and adversarially trained model, these images are almost indistinguishable from the target while retaining high model confidence for the source class. Since adversarially robust models have

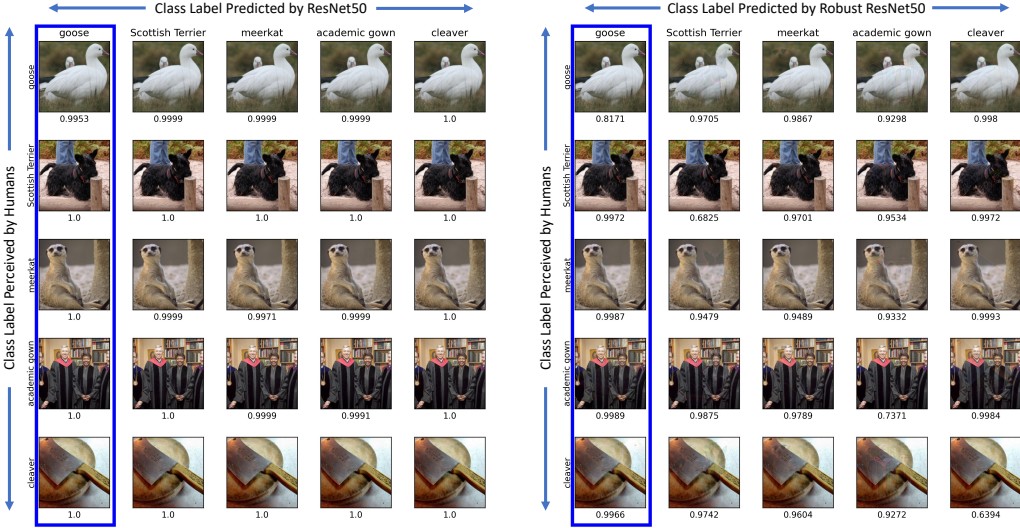

Figure 3: The images returned by LST for 5 random source and target images for normally trained (left) and adversarially trained (right) ResNet-50 models. The image in the $i^{th}$ row and $j^{th}$ column is the LST output obtained using the $i^{th}$ image as target and $j^{th}$ image as the source. The confidence of the model prediction for the source class (names on top of each column) is displayed just below each image. For example, all four LST output blindspots in the first column (highlighted), using the source 'goose' image, are all predicted to be of the 'goose' class with very high confidence. Diagonal images are unchanged, as source equals target. We observe that almost any source image can be iteratively modified using LST to resemble any target image very closely without any loss in confidence for both normal and adversarially trained ResNet-50 models.

perceptually aligned gradients, we can sometimes visually notice a few traces of the source image in the final LST image; for example the 'meerkat' image in the 3rd row, 2nd column in the right side of Fig 3 has some traces of the source 'terrier' image, but differences are usually hard to perceive.

We also examine the model confidence over linear interpolations between the source image and LST outputs for all target pairs in Fig 6. Formally, consider a source image $x_s$ with label $y$ and let $x_{op}$ represent the LST output when applied toward a target image $x_t$. Denote the difference vector as $\Delta v = x_{op} - x_s$. Then, we observe that $x_s + \alpha \Delta v$ is assigned high confidence with respect to class $y$ by the model $\forall \alpha \in [0, 1]$, which represents the entire linear interpolant path between $x_s$ and $x_{op}$. Furthermore, we observe that the path discovered by the Level Set Traversal algorithm enjoys two key properties: (1) Uniqueness (once the target image is fixed) and (2) Extremality:

(1) *Uniqueness:* Since the local tangent space of the level set is $(d-1)$ dimensional, several independent directions are orthogonal to the local gradient, and apriori do not yield a unique path like a gradient-flow. However, once we fix our target image, we use its difference vector with respect to the current iterate (L4) and compute its projection onto the local tangent space (L7) of the level set, thereby generating a *uniquely defined path*.

(2) *Extremality:* Though this flow-based path may be non-linear, we additionally discover that the final output-point of this flow is surprisingly linearly connected with high-confidence to the source image after we apply discretized approximations in practical settings for common vision models etc. Formally, the LST output $x_{op}$ is *linearly extremal* in the sense that $x_s + (1+\epsilon)\Delta v$ is rapidly assigned low-confidence by the model even for extremely small values of $\epsilon > 0$, where $\Delta v = x_{op} - x_s$.

Thus, using the LST algorithm, we find that the level sets of common models extend outwards in an expansive, connected manner to include images of *arbitrary* classes, which human oracles would never state as being similar. Since the linear path from any given source image to LST outputs for arbitrary target images retains high model confidence throughout, it unveils a remarkable star-like substructure for superlevel sets as shown in Fig 4, where the number of "limbs" or linear protuberances of the star-like structure is *extraordinarily large*, plausibly as large as the number of images in all other classes. Furthermore, to study the size of the level sets beyond the one-dimensional interpolant paths, we analyze the two-dimensional triangular convex hull between a given source image and two LST output blindspot images, using quantitative metrics in Section 6.

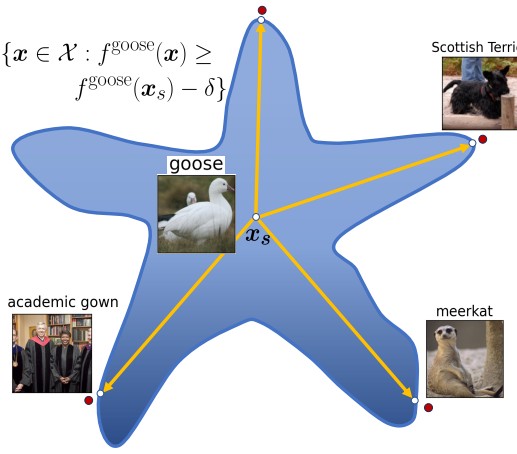

$\{x \in \mathcal{X} : f^{\text{goose}}(x) \geq f^{\text{goose}}(x_S) - \delta\}$

Scottish Terrier

goose

$x_s$

academic gown

meerkat

Figure 4: Schematic of Star-like set substructure of Superlevel sets: The linear interpolant paths between the source image $x_s$ and blindspots found using LST maintain high-confidence throughout for arbitrary target images of other classes.

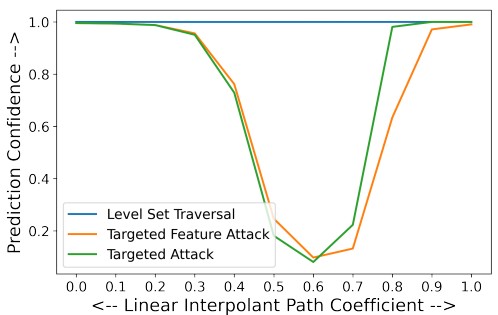

Figure 5: Model Confidence over Linear Interpolant Paths: We plot the median prediction confidence with respect to class $y_2$ over linear paths between the target benign sample ($x_2$) and adversarial examples (such as $x_1 + \varepsilon_{12}$ targeted towards input $x_2$) in input space. While model confidence falls to near-zero using standard targeted adversarial attacks, the linear path between the LST output and source image has high confidence of almost 1.0 throughout.

## 4    Disconnected Nature of Standard Adversarial Attacks

At first glance, the images output by the LST algorithm may seem similar to those produced using targeted variants of standard adversarial attacks. In this section, we explore the connectivity of high-confidence paths as induced by standard adversarial attacks, and show that adversarially attacked source images are not linearly connected with high-confidence paths to their target image. In detail, let $(x_1, y_1)$ and $(x_2, y_2)$ be any two data samples from different classes ($y_1 \neq y_2$). A targeted adversarial attack [Carlini and Wagner, 2017] on input $x_1$ with respect to class $y_2$ is formulated as solving $\varepsilon_{12} = \arg\min_{\varepsilon_{12}:||\varepsilon_{12}|| \leq \epsilon} CE(f(x_1 + \varepsilon_{12}), y_2)$. A related variant, called a feature-level targeted adversarial attack [Sabour et al., 2015] instead uses intermediate network activations to craft image perturbations. If $f_{|L}(x)$ represents the network activations from hidden layer $L$ for an input $x$, then this attack attempts to match features by optimizing $\varepsilon_{12} = \arg\min_{\varepsilon_{12}:||\varepsilon_{12}|| \leq \epsilon} ||f_{|L}(x_1 + \varepsilon_{12}) - f_{|L}(x_2)||$. The hidden layer ($L$) that is often selected corresponds to pre-activations of the final fully-connected layer, and thus often induces misclassification.

Using this framework, we can thus analyze the connectivity of high-confidence paths with respect to class $y_2$ between the target benign sample ($x_2$) and adversarial examples (such as $x_1 + \varepsilon_{12}$ targeted towards $x_2$), using the linear interpolant path between the two. By doing so, we can analyze the level set with respect to class $y_2$ using targeted adversarial examples alone, though the perturbation utilized is inherently norm-limited. In Fig 5, we plot the median confidence of a normally trained ResNet-50 model over the linear interpolant paths for 1000 source-target pairs. We find that though the model confidence with respect to class $y_2$ is high for both the target benign input $x_2$ and the targeted adversarially attacked input $x_1 + \varepsilon_{12}$ (that is, at the end-points of the linear path), the model *does not* maintain high confidence over the linear interpolant path between the two, but sharply declines to a valley of near zero-confidence over the path. For specific inputs, we often observe sharp, erratic diminutions in target-class prediction confidence over the convex combination, with a non-trivial region of near-zero confidence. This contrasts sharply with the existence of linear-connectivity observed between a source image and LST blindspot images. We thus crucially note that adversarial attacks are *standalone insufficient* to study the structure of level sets of common vision models. However, we incorporate them into our proposed Level Set Traversal algorithm in a fruitful manner.

## 5    Theoretical Analysis for Common Models

To better understand the geometry of level set submanifolds for a given prediction confidence threshold, we analyze the nature of confidence preserving perturbations in a few simplified model settings.

**1) Linear Functional:** First, consider the classification model to be a linear functional $f : \mathbb{R}^d \to \mathbb{R}$, that is, $f(\boldsymbol{x}_1 + \boldsymbol{x}_2) = f(\boldsymbol{x}_1) + f(\boldsymbol{x}_2)$ and $f(\lambda \boldsymbol{x}) = \lambda f(\boldsymbol{x})$. Then, by the Riesz Representation theorem, there exists a unique vector $\boldsymbol{w}_f \in \mathbb{R}^d$ such that $f(\boldsymbol{x}) = \langle \boldsymbol{w}_f, \boldsymbol{x} \rangle \ \forall \boldsymbol{x} \in \mathbb{R}^d$. We thus observe that for any vector $\boldsymbol{v}$ orthogonal to $\boldsymbol{w}_f$, $f(\boldsymbol{x} + \boldsymbol{v}) = \langle \boldsymbol{w}_f, \boldsymbol{x} + \boldsymbol{v} \rangle = \langle \boldsymbol{w}_f, \boldsymbol{x} \rangle + \langle \boldsymbol{w}_f, \boldsymbol{v} \rangle = \langle \boldsymbol{w}_f, \boldsymbol{x} \rangle$. Thus, in this setting, the level sets of $f$ are $(d-1)$ dimensional *linear subspaces* spanned by the set $\{ \boldsymbol{v} \in \mathbb{R}^d : \langle \boldsymbol{w}, \boldsymbol{v} \rangle = 0 \}$. We observe that a similar argument can be extended to affine functions of the form $f(\boldsymbol{x}) = \langle \boldsymbol{w}_f, \boldsymbol{x} \rangle + c$, where $c \in \mathbb{R}$.

We remark that by applying a first-order Taylor series approximation for general real-valued smooth functions, we observe near-affine behavior locally within a small neighborhood: $f(\boldsymbol{x} + \epsilon \boldsymbol{a}) = f(\boldsymbol{x}) + \epsilon \langle \nabla f(\boldsymbol{x}), \boldsymbol{a} \rangle + O(\epsilon^2 \|\boldsymbol{a}\|^2)$. Thus locally, we observe that confidence-preserving perturbations can arise from a $(d-1)$ dimensional plane orthogonal to the gradient. Indeed we incorporate this implicitly in our proposed Level Set Traversal algorithm, wherein the orthogonal projection $\Delta \boldsymbol{x}_\perp$ with respect to the local gradient is iteratively computed, and its relative success can partly be attributed to the orthogonal hyperplane having a large dimension, namely $(d-1)$.

Another related setting worthy of note is that of neural networks that utilize ReLU activations, which induces a piece-wise linear structure over tessellated subsets of the input domain. Thus, a given output neuron of a ReLU network functionally has the form $f(\boldsymbol{x}) = \langle \boldsymbol{w}, \boldsymbol{x} \rangle + c$ within a given tessellated region, and thus has a constant gradient within the same region. Further, between two such adjacent regions, the two orthogonal $(d-1)$ dimensional hyperplanes typically intersect over an affine space over dimension at least $(d-2)$. Thus if $\boldsymbol{x}_1, \boldsymbol{x}_2$ are two inputs such that their linear interpolant path cuts across $n$ distinct tessellated regions, the common intersection of these orthogonal hyperplanes will typically be $(d-n)$ dimensional, indicating that there exist perturbations which lie in the common null-space of the gradients as defined along each tessellated region that is crossed. Indeed in the following section, we demonstrate empirically for Residual Networks that though the exact iterative path followed by the Level Set Traversal algorithm is discretized and non-linear, the final outputs so found are often linearly connected through paths of high confidence to the source image, thereby lying within the level set of the original source class. This indicates that the overlap of different $(d-1)$ dimensional hyperplanes is non-trivial at a non-local scale in image space, whereby we observe extended connected regions of high confidence.

**2) Full-Rank Linear Transformations:** Let us now consider a setting apart from classification, such as regression, wherein the complete vector representation of the output is of principal interest. Let the model be of the form $f : \mathbb{R}^d \to \mathbb{R}^d$, $f(\boldsymbol{x}) = A\boldsymbol{x}$, where $A \in \mathbb{R}^{d \times d}$ is a full-rank matrix. In this setting, we observe that if $f(\boldsymbol{x}_1) = f(\boldsymbol{x}_2) = A\boldsymbol{x}_1 = A\boldsymbol{x}_2$, implying that $A(\boldsymbol{x}_1 - \boldsymbol{x}_2) = \boldsymbol{0}$, the zero-vector. But since $A$ is full-rank, this implies that $\boldsymbol{x}_1 - \boldsymbol{x}_2 = \boldsymbol{0}$, i.e, $\boldsymbol{x}_1 = \boldsymbol{x}_2$. Thus, $f$ is a bijective function and has a trivial null space. Thus in this setting, we necessarily have to relax the problem to be that of identifying perturbations of large magnitude to the input that *minimally* change the function output: that is, we solve for $\min_{\boldsymbol{v} \neq \boldsymbol{0}} \frac{\|A\boldsymbol{v}\|^2}{\|\boldsymbol{v}\|^2}$. Indeed, let the Singular Value Decomposition (SVD) of the full rank matrix $A$ be given by $A = U\Sigma V^T$, where $U, V$ are $d \times d$ orthogonal matrices, and $\Sigma$ is a diagonal matrix consisting of the positive singular values $\sigma_i$ for $1 \leq i \leq d$, in descending order without loss of generality. Then, $\|A\boldsymbol{v}\| = \|U\Sigma V^T \boldsymbol{v}\| = \|U(\Sigma V^T \boldsymbol{v})\| = \|\Sigma V^T \boldsymbol{v}\|$ since $U$ is an orthogonal matrix. Similarly, since $V$ is orthogonal as well, let $\boldsymbol{z} = V^T \boldsymbol{v}$, so that $\|\boldsymbol{z}\| = \|\boldsymbol{v}\|$. Thus, $\min_{\boldsymbol{v} \neq \boldsymbol{0}} \frac{\|A\boldsymbol{v}\|^2}{\|\boldsymbol{v}\|^2} = \min_{\boldsymbol{z} \neq \boldsymbol{0}} \frac{\|\Sigma \boldsymbol{z}\|^2}{\|\boldsymbol{z}\|^2} = \min_{\boldsymbol{z}} \|\Sigma \boldsymbol{z}\|^2$ such that $\boldsymbol{z}^T \boldsymbol{z} = 1$. But since $\Sigma$ is diagonal, $\min_{\boldsymbol{z}} \|\Sigma \boldsymbol{z}\|^2 = \sum_{i=1}^d \sigma_i^2 z_i^2$, under the constraint that $||z||^2 = 1$. It is then easy to observe that the minimum value attained is $\sigma_d^2 = \sigma_{min}^2$, and is attained when the input vector to $f$ is the right-singular vector corresponding to the minimum singular value of $A$.

In this setting, we remark that adversarial perturbations, in contrast, can be formulated in this setting as $\max_{\boldsymbol{v} \neq \boldsymbol{0}} \frac{\|A\boldsymbol{v}\|^2}{\|\boldsymbol{v}\|^2}$, with the maximum value given by $\sigma_{max}^2$ and attained by the right-singular vector corresponding to the maximum singular value of $A$, highlighting the complementary nature of the two problems as indicated previously in Section 2.1. Indeed, the condition number $\kappa$ of an invertible matrix $A$ is defined as $\kappa = \sigma_{max}/\sigma_{min} = \|A\| \cdot \|A^{-1}\|$. If condition number is large with $\kappa \gg 1$, the matrix $A$ is said to be ill-conditioned, and induces an inescapable compromise: if $\sigma_{max} = \|A\| \approx 1$ (as potentially expected in "robust" networks), then $1/\sigma_{min} = \|A^{-1}\| \gg 1$ is necessarily large, thereby inducing extreme *under-sensitivty* along some dimensions; while if $1/\sigma_{min} = \|A^{-1}\| \approx 1$, then $\sigma_{max} = \|A\| \gg 1$ and the model is extremely *over-sensitive*, similar to the phenomenon of adversarial vulnerability.

Table 1: Quantitative image distance metrics between output of Level Set Traversal and target images.

| Models | RMSE : $\mu \pm \sigma$ | $\ell_\infty$ dist: $\mu \pm \sigma$ | SSIM: $\mu \pm \sigma$ | LPIPS dist: $\mu \pm \sigma$ |
|---|---|---|---|---|
| ResNet-50 (Normal) | $0.008 \pm 0.001$ | $0.046 \pm 0.020$ | $0.990 \pm 0.021$ | $0.002 \pm 0.004$ |
| ResNet-50 (AT) | $0.029 \pm 0.008$ | $0.746 \pm 0.124$ | $0.915 \pm 0.041$ | $0.057 \pm 0.037$ |
| DeiT-S (Normal) | $0.011 \pm 0.002$ | $0.116 \pm 0.030$ | $0.973 \pm 0.024$ | $0.024 \pm 0.017$ |
| DeiT-S (AT) | $0.046 \pm 0.010$ | $0.821 \pm 0.117$ | $0.898 \pm 0.041$ | $0.219 \pm 0.068$ |

## 6 Quantifying Under-Sensitivity of Vision Models

In this paper, we primarily consider standard vision datasets such as ImageNet [Deng et al., 2009] and CIFAR-10 [Krizhevsky et al., 2009] (latter in Section C of the Appendix). We thereby explore the "blind-spots" of popular vision models such ResNet [He et al., 2016] and Vision Transformers [Dosovitskiy et al., 2020, Touvron et al., 2021]. Furthermore, we explore the connectivity of such level sets on normally trained variants of such networks, as well as adversarially robust counterparts. For the latter, we analyze robust models trained adversarially against $(4/255)$ $\ell_\infty$ constrained adversaries, available on RobustBench [Croce et al., 2021]. Specifically, we utilize a robust ResNet-50 model from Salman et al. [2020] and a robust DeiT model from Singh et al. [2023]. We also fix the hyperparameters of LST for all models for a fair comparison (detailed in Section E of the Appendix).

**Image Quality Metrics:** We now proceed to quantitatively verify our observations in Section 3. First, to help quantify the deviation between target images and blindspot outputs generated by our proposed Level Set Traversal algorithm, we utilize a combination of classical image metrics such as RMSE, $\ell_\infty$ and Structural Similarity Index (SSIM), and perceptual measures such as LPIPS distance using AlexNet [Zhang et al., 2018]. For SSIM, a higher value indicates a closer match, while for all other metrics, a lower value indicates a closer match. To calculate these metrics, we sample around 1000 source images from ImageNet, and select five other random target images of different classes for each source image. We present the image quality metrics for blindspots discovered by LST in Table 1. Here the standard deviation is over the different randomly chosen source and target images.

**Metrics for Model Confidence:** To evaluate the extent of the model invariance over the regions between the LST outputs and the source image, we evaluate the model confidence with respect to the source class over one-dimensional linear interpolant paths and over two-dimensional subspaces as well. For the latter, we evaluate the model confidence over the triangular convex hull obtained by linear interpolation over three reference points, namely the source image and the two target blindspot images produced using LST. For example, the input at the 'centroid' of the triangle formed by a source image and pair of target blindspots is the arithmetic mean of the three images. We visualize these in Fig 6, wherein the prediction confidence (in the range $[0, 1]$) assigned by the model with respect to the source class is mapped to a continuous colorbar, with high-confidence points (close to 1.0) appearing as bright yellow, and low-confidence points (close to 0.0) appearing as dark violet. Specifically, we use the following metrics: (a) **Average Triangle ($\triangle$) Confidence**: the mean of the model's source class confidence over the enclosed triangle, (b) **Average Triangle ($\triangle$) Fraction** for various values of $\delta$: the fraction of inputs in the triangular region for which the model confidence is greater than $p_{\text{src}} - \delta$, averaged over all possible target blindspot pairs, where $p_{\text{src}}$ is the confidence of the source image, (c) **Average Path Confidence**: the average model confidence over all linear paths from the source image to all LST blindspot images. The higher these metrics, the more confident, and thus invariant, the model is in this region. For computing these metrics, we use linear interpolations between the source images and the 5 LST outputs found previously for computing the distance metrics. We thus use $\binom{5}{2} = 10$ triangles for each source image, and sample this triangular area in an equispaced manner to obtain 66 images for computation of the triangle ($\triangle$) metrics. We present these metrics in Table 2, along with the mean (and standard deviation) of model confidence on the source class images ($p_{\text{src}}$) for reference. Here, the standard deviation is over the different randomly chosen source images.

We can now quantitatively confirm many of the trends we qualitatively observed in Section 3. In Table 1, we observe that the LST outputs found for the models are closer to the targets as compared to the adversarially trained models. The difference is particularly stark when comparing the $\ell_\infty$ distances, as $\ell_\infty$ is a particularly sensitive metric and is also the threat model against which the models were trained. However, for other distance metrics like RMSE or LPIPS, the difference between normally trained and adversarially trained ResNet-50 is not as high. In particular, LPIPS

Table 2: Quantitative confidence metrics over the triangular convex hull ($\Delta$) of a given source image and two target LST blindspot image-pairs and over linear interpolant paths between source and blindspot images. (For reference, a random classifier would have confidence of 0.001)

| Models | $p_{\text{src}}$ ($\mu \pm \sigma$) | Avg $\Delta$ Conf. ($\mu \pm \sigma$) | Avg $\Delta$ Frac. ($\mu \pm \sigma$) | | | | Avg Path Conf. ($\mu \pm \sigma$) |
|---|---|---|---|---|---|---|---|
| | | | $\delta = 0.0$ | $\delta = 0.1$ | $\delta = 0.2$ | $\delta = 0.3$ | |
| ResNet-50 (Normal) | $0.99 \pm 0.02$ | $0.56 \pm 0.10$ | $0.13 \pm 0.15$ | $0.51 \pm 0.11$ | $0.53 \pm 0.1$ | $0.54 \pm 0.10$ | $0.96 \pm 0.05$ |
| ResNet-50 (AT) | $0.88 \pm 0.11$ | $0.83 \pm 0.09$ | $0.49 \pm 0.29$ | $0.79 \pm 0.13$ | $0.85 \pm 0.1$ | $0.88 \pm 0.09$ | $0.93 \pm 0.06$ |
| DeiT-S (Normal) | $0.85 \pm 0.06$ | $0.68 \pm 0.05$ | $0.54 \pm 0.11$ | $0.67 \pm 0.06$ | $0.71 \pm 0.06$ | $0.73 \pm 0.06$ | $0.94 \pm 0.02$ |
| DeiT-S (AT) | $0.76 \pm 0.08$ | $0.59 \pm 0.07$ | $0.20 \pm 0.09$ | $0.43 \pm 0.14$ | $0.63 \pm 0.15$ | $0.76 \pm 0.12$ | $0.73 \pm 0.06$ |

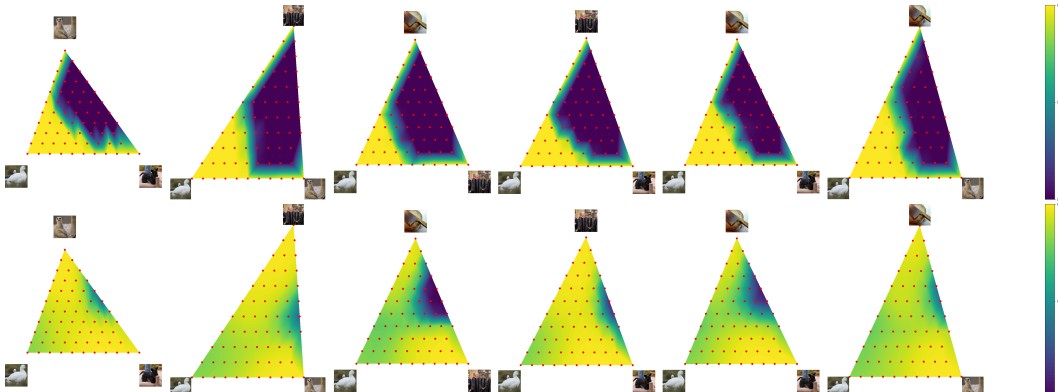

Figure 6: Visualization of confidence of standard (top) and robust (bottom) ResNet-50 models over the triangular convex hull of a source 'goose' image and two LST outputs for all pairs of target images from 4 other classes (same images as in Fig 3). In all source-target image pairs, the linear interpolant path maintains high confidence, implying that the source image is linearly connected to the LST target output in a star-like substructure within the level set. For adversarially trained models, we observe that a significant fraction of the triangular hull lies in the superlevel sets of high-confidence, thereby indicating their under-sensitivity in regions far beyond their original threat model.

distances for both models are low, which indirectly implies that the perceived difference between the images for humans oracles is relatively low. This can also be confirmed by visually inspecting the images in Fig 3. For the normally trained DeiT-S, the distance metrics are very similar to that of ResNet-50, with slightly higher RMSE and LPIPS distances. However, the adversarially trained (AT) variant is significantly less under-sensitive compared to both its normal counterpart and the ResNet-50 (AT). Specifically, the LPIPS distance metric is much greater for DeiT-S (AT), which implies there exist significant human-perceptible differences in the image. We can confirm this in Fig 7, where the LST output for the adversarially trained DeiT-S model contains visible traces of the source image. However, the LST outputs are still clearly much closer to the target class as compared to the source, which indicates that there are still some significant blind spots in DeiT-S (AT).

However, when we measure the extent of model invariance over the convex regions enclosed between the LST output and the target images, we find that adversarially trained ResNet-50 are overall *more* invariant (or under-sensitive) as compared to the normally trained variant. For example, the average triangle confidence for ResNet-50 (AT) is higher than that of normally trained ResNet-50, even though its source confidence is much lower. We also find that a much larger fraction of the triangular convex hull lies within the superlevel set for $p_{\text{src}} - \delta$ for ResNet-50 (AT) as compared to normal ResNet-50 for all values of $\delta$. The average path confidence is much closer to $p_{\text{src}}$ for ResNet-50 (AT) as compared to normally trained ResNet-50. This quantitatively verifies the observation made in Fig 6 that adversarial training demonstrably exacerbates under-sensitivity. Interestingly, these robust models are under-sensitive over subsets of the input domain that lie well beyond the original threat model used in its training. Moreover, between the normally trained DeiT-S and ResNet-50 models, the former appears to be more invariant with greater average confidence over the triangular convex hulls, despite having lower source image confidences $p_{\text{src}}$. For the robust variant of DeiT-S however, the trend is less apparent due to the significantly lower average source image confidences $p_{\text{src}}$. However the average relative $\Delta$ fraction becomes higher for larger values of $\delta$ (such as 0.3), indicating that the superlevel sets are indeed expansive, albeit for lower confidence thresholds.

# 7 Discussion

While the existence of level sets alone is not very significant in itself, using the proposed LST algorithm, we find that the level set for common vision models is remarkably expansive — large enough to contain inputs that look near-identical to arbitrary target images from other classes. Since the linear path from any given source image to LST blindspot outputs retain high model confidence throughout, the level sets have a star-like connected substructure, where the number of 'limbs' or linear protuberances of the star-like structure is *extraordinarily large*, plausibly as large as the number of images in all other classes. This is considerably noteworthy since it indicates the hitherto unknown and unappreciated scale and extent of under-sensitivity in common vision models. Moreover this hints at the potential degree of difficulty towards adequately mitigating this phenomenon in practical settings. For instance, if the level set for images of class $y_1$ contained sizable protuberances towards only one other class $y_2$ alone, the problem could perhaps be tackled by introducing a contrastive training objective that encourages the network to better discriminate between $y_1 - y_2$ image pairs by utilizing a denser sampling of related image augmentations, likely resulting in the diminution of these specific 'directed' protuberances (assuming reasonable train-test generalization). But since the star-like connected substructure uncovered by LST implies that such protuberances exist towards any generic image of any other class, such simple approaches will likely be ineffective and possibly computationally infeasible from a combinatorial perspective. Thus, based on the observations uncovered with LST, we hypothesize that addressing the pervasive issue of under-sensitivity in conventional vision models might present a significantly non-trivial challenge.

# 8 Related Work

The phenomenon of under-sensitivity in classification models was first pointed out by Jacobsen et al. [2018a], wherein they utilize a class of invertible neural networks called fully Invertible RevNets [Jacobsen et al., 2018b, Kingma and Dhariwal, 2018] to specifically craft input images that do not affect network activations at a given layer. In contrast, our proposed algorithm is applicable to general network architectures since we solely utilize input-gradients to perform the traversal over image space. Tramèr et al. [2020] further demonstrated that due to the misalignment of $\ell_p$ norm bounded balls and the ideal set of human-imperceptible perturbations, networks that are adversarially trained against such $\ell_p$ bounded perturbations of relatively large radius are overly-smooth, and become excessively susceptible to invariance-based attacks within the same $\ell_p$ radius. To find such images that lie within a given $\ell_p$ threat model, but induce human oracles to change their label assignment, the authors propose to identify the training image from another class closest in image space and apply a series of semantic-preserving transformations, and additionally use techniques such as realignment and spectral clustering. Given that these operations are fairly complex, the attack algorithm is slow, with alignment alone requiring an amortized time of minutes per input example. In contrast, our proposed method relies upon gradient-backpropagation steps which are efficiently parallelized across a minibatch of input images. Furthermore, our technique is seen to be successful for arbitrary source-target image pairs, since we do not utilize near-neighbour training images as the target. On the theoretical front, Jayaram et al. [2020] analyze the problem of span-recovery of neural networks given only oracle access to the network predictions. They characterize the feasibility of span recovery, and thus approximations of the null space of networks in a provable setting, but remark that its success in practical settings is potentially limited to networks that are extremely thin.

# 9 Conclusions

In this work, we investigate the phenomenon of under-sensitivity of classification models, wherein large-magnitude semantic perturbations leave network activations unchanged. To identify such "blind spots" that occur within high-confidence level sets of common vision models, we develop a novel Level Set Traversal algorithm that iteratively perturbs a given source image to appear visually similar to an arbitrary target image by utilizing orthogonal projections with respect to the local input gradient. The proposed method is applicable to general neural networks, and helps uncover a star-like substructure for the level and superlevel sets of CNNs and ViTs on common datasets. We further observe that adversarially trained models retain a high-degree of confidence over regions that lie far beyond its original threat model, with super-level sets that extend over a significant fraction of the triangular convex hull between a given source image and arbitrary pair of blindspot images.

## 10 Acknowledgements

This project was supported in part by a grant from an NSF CAREER AWARD 1942230, ONR YIP award N00014-22-1-2271, ARO's Early Career Program Award 310902-00001, Meta grant 23010098, HR00112090132 (DARPA/RED), HR001119S0026 (DARPA/GARD), Army Grant No. W911NF2120076, NIST 60NANB20D134, the NSF award CCF2212458, an Amazon Research Award and an award from Capital One.

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

# Appendix

## A  Theoretical Characterization of Level Sets

Let $g : \mathbb{R}^d \to \mathbb{R}$ be a differentiable function. For $c \in \mathbb{R}$, the level set $L_g(c)$ is said to be *regular* if the gradient of $g$ is non-vanishing for all points in the level set, that is, $\nabla g(\boldsymbol{x}) \neq \boldsymbol{0} \ \forall \boldsymbol{x} \in L_g(c)$. We can then show that the regular level sets of continuously differentiable functions are $(d-1)$ dimensional submanifolds of $\mathbb{R}^d$ (Lemma 1), using the Implicit Function Theorem. We restate the lemma here for the readers' convenience, and present its proof below.

**Lemma 1.** *If $g : \mathbb{R}^d \to \mathbb{R}$ is a continuously differentiable function, then each of its regular level sets is a $(d-1)$ dimensional submanifold of $\mathbb{R}^d$.*

*Proof.* Without loss of generality, let $L_g(0)$ be a regular level set of $g$, and let $\boldsymbol{a} \in L_g(0)$ (if $c \neq 0$, we can define a new function shifted by the constant $-c$, so it suffices to consider $L_g(0)$). By definition of regularity, we know that $\nabla g(\boldsymbol{a}) \neq \boldsymbol{0}$, and at least one of its components must be non-zero. By renumbering the coordinates, without loss of generality, we can assume the first vector component is non-zero, i.e. $\nabla^1 g(\boldsymbol{a}) = \frac{\partial}{\partial x_1} g(\boldsymbol{a}) \neq 0$. Thus we have a function $g$ such that $g : \mathbb{R} \times \mathbb{R}^{d-1} \to \mathbb{R}$ with

$$g(a_1, a_2, \ldots, a_d) = 0 \quad , \quad \frac{\partial g}{\partial x_1}(a_1, a_2, \ldots, a_d) \neq 0$$

Since the $1 \times 1$ Jacobian submatrix $\left[ \frac{\partial}{\partial x_1} g(\boldsymbol{a}) \right]$ is non-zero and thus invertible, we can thus apply the Implicit Function Theorem to $g$ at $\boldsymbol{a}$, which states that there exist open sets $U, V$ such that $U \subset \mathbb{R}$ with $a_1 \in U$ and $V \subset \mathbb{R}^{d-1}$ with $(a_2, \ldots, a_d) \in V$, and an "implicit" continuously differentiable function $t : V \to U$ such that $t(a_2, \ldots, a_d) = a_1$ with the property that for all $(x_2, \ldots, x_d) \in V$, we have:

$$g(t(x_2, \ldots, x_d), x_2, \ldots, x_d) = 0$$

Further, we observe that the set $W = U \times V \subset \mathbb{R}^d$ is open since $U, V$ are open. Now, if $\boldsymbol{x} \in L_g(0) \cap W$, $x_1 = t(x_2, \ldots, x_d)$. Thus, consider $\phi_{\boldsymbol{a}} : L_g(0) \cap W \to \mathbb{R}^d$ with

$$\phi_{\boldsymbol{a}}(x_1, x_2, \ldots, x_d) = (g(t(x_2, \ldots, x_d), x_2, \ldots, x_d), x_2, \ldots, x_d)$$

which is a map with its first coordinate identically zero. Thus, if $\Pi_{-1}(x_1, x_2, \ldots, x_d) = (x_2, \ldots, x_d)$ is the projection map from $\mathbb{R}^d$ to $\mathbb{R}^{d-1}$ that omits the first coordinate, $\Pi_{-1} \circ \phi_{\boldsymbol{a}}$ is a continuously differentiable chart from $L_g(0) \cap W$ to $\mathbb{R}^{d-1}$ containing $\boldsymbol{a}$. Since this is true pointwise for each $\boldsymbol{a} \in L_g(0)$, $L_g(0)$ is a $(d-1)$ dimensional submanifold of $\mathbb{R}^d$. $\qquad\square$

## B  Implementation Details and Methodology

### B.1  Details on Datasets and Pretrained Models

In this paper, we present results on vision datasets such as ImageNet [Deng et al., 2009] and CIFAR-10 [Krizhevsky et al., 2009], given that they have come to serve as benchmark datasets in the field. ImageNet Large Scale Visual Recognition Challenge (ILSVRC) consists of over a million labelled RGB color images arising from a wide variety of objects, totalling to 1000 distinct classes. The images themselves are of relatively high resolution, with common vision models utilizing an input shape of $224 \times 224$ pixels. On the other hand, CIFAR-10 is a smaller dataset, consisting of $32 \times 32$ pixel RGB images from ten classes: "airplane", "automobile", "bird", "cat", "deer", "dog", "frog", "horse", "ship" and "truck". In this paper, all training and experimental evaluations were performed using Pytorch [Paszke et al., 2019].

The $\ell_\infty$ norm constrained threat model is the most commonly used setting for adversarial robustness in both these datasets. On ImageNet, we analyse robust models trained with adversaries constrained within a radius of $4/255$, while on CIFAR-10, we consider robust models trained with adversaries over a larger radius of $8/255$. We use adversarially trained models as made available on RobustBench [Croce et al., 2021] to analyze popular vision models. For the class of Residual Networks [He et al., 2016], we analyze a normally trained ResNet-50 model from the Pytorch Image Models (timm) repository, and the adversarially robust ResNet-50 model from Salman et al. [2020]. For the class of Vision transformers [Dosovitskiy et al., 2020, Touvron et al., 2021], we utilize a robust DeiT model from Singh et al. [2023]. For CIFAR-10, we utilize adversarially trained WideResNet-34-10 models

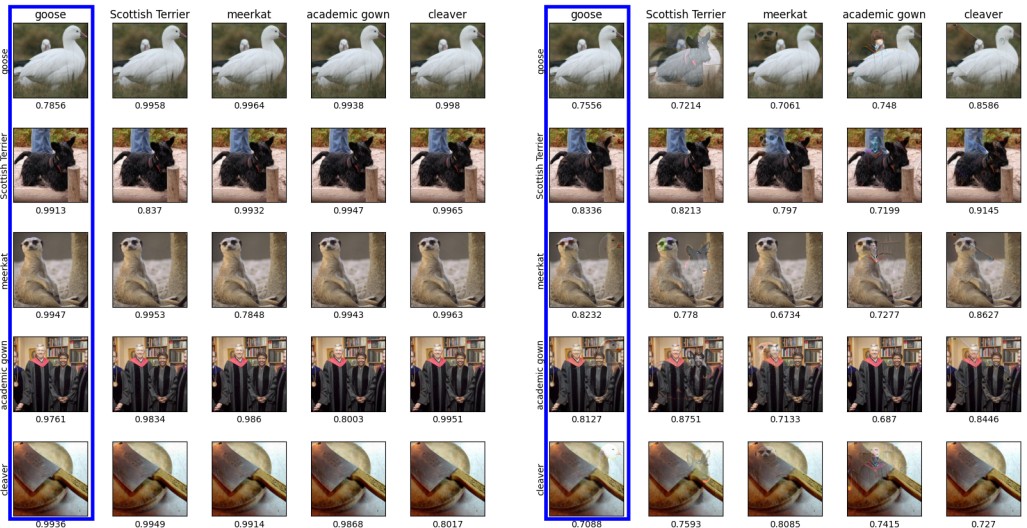

Figure 7: The images returned by LST for the same 5 source and target images as in 3 for normally trained (left) and adversarially trained (right) DeiT-S models. We again observe the LST outputs are highly similar to the target images in this setting as well.

from Wu et al. [2020], Zhang et al. [2019b] and Rade and Moosavi-Dezfooli [2022], alongside a normally trained counterpart.

### B.2 Algorithm and Hyperparameter Details

**ImageNet** : In the main paper, we fix the parameters of the LST algorithm for all the visualizations (Fig 3,6,7 and Tables 1,2 in the Main paper). The scale factor for the step perpendicular to the gradient, or $\eta$, is $10^{-2}$. The stepsize for the perturbation parallel to the gradient $-\nabla CE(f(\boldsymbol{x}), y)$, or $\epsilon$, is $2 \times 10^{-3}$. The confidence threshold ($\delta$) is 0.2, which means that the confidence never drops below the confidence of the source image by more than 0.2. In practice, we rarely observe such significant drops in the confidence during the level set traversal. The algorithm is run for $m = 400$ iterations. The number of iterations required is relatively high compared to standard adversarial attacks because of the much larger distance between the source and the target images. We also present results with different parameters in the subsequent sections.

**CIFAR-10** : We fix the parameters of LST when applied on CIFAR-10 in a similar manner. The scale factor for the step perpendicular to the gradient, or $\eta$, is $1 \times 10^{-2}$. The stepsize for the perturbation parallel to the gradient $-\nabla CE(f(\boldsymbol{x}), y)$, or $\epsilon$, is $2 \times 10^{-3}$. The confidence threshold ($\delta$) is 0.25, and we run the algorithm for $m = 300$ iterations to obtain high fidelity output images. The adversarially trained model is observed to have low prediction confidence even on the source images, and thus again requires a large number of iterations for good convergence.

## C Results on CIFAR-10

In this section, we present qualitative and quantitative results obtained on the CIFAR-10 dataset. Similar to Section 6 of the Main paper, we consider RMSE, $\ell_\infty$, the Structural Similarity Index (SSIM), and LPIPS with AlexNet [Zhang et al., 2018]. For SSIM, a higher value indicates a closer match, while for all other metrics, a lower value indicates a closer match. To calculate these metrics, we sample at 1000 images from CIFAR-10 to serve as the source images, and consider 9 fixed images from all the remaining 9 classes as targets as shown in Figure-8. By visual inspection, we see that any source image can be iteratively modified by LST to resemble any given target image, while retaining high confidence on the original class as indicated in each column of the figure. Indeed, we quantify this in Table-3 where we observe that the final output images of the LST traversal algorithm are close

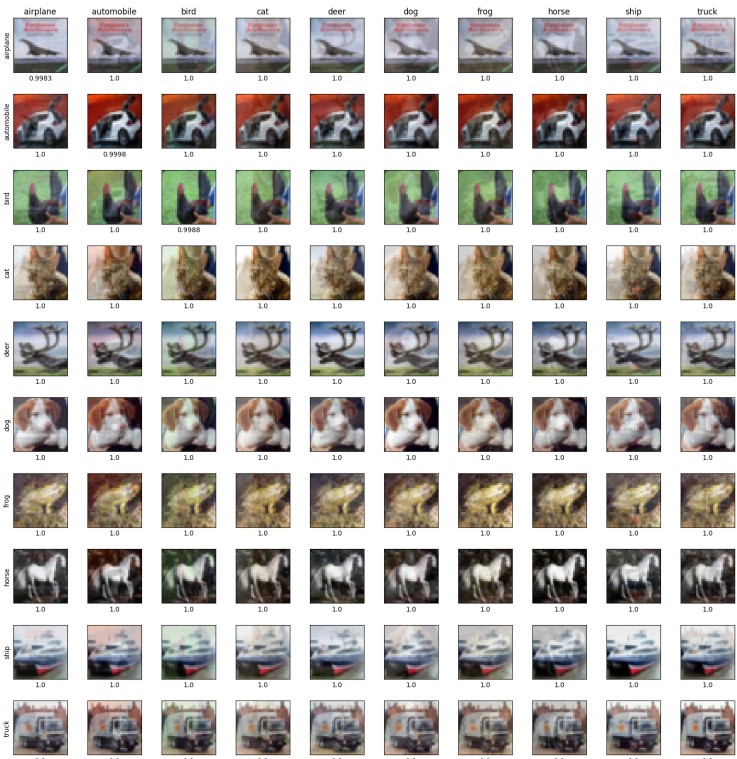

Figure 8: CIFAR10 dataset- Images returned by the Level Set Traversal algorithm for a normally trained WideResNet-28 model.The image in the $i$-th row and $j$-th column is obtained by using the $j$-th image as the source image and $i$-th image as target. The images in the diagonal are the unchanged random images, as the source and target are identical. The confidence of the model prediction for the source class (names on top of each column) is displayed just below each image. We observe that almost any source image can be iteratively changed to resemble any target image very closely without any loss in confidence, as indicated by images presented in any given column.

to the target over these different metrics; in particular the high SSIM similarity and small LPIPS distance implies high visual similarity for human oracles.

Similar to the Main paper, we again evaluate the extent of the model invariance over the regions between the LST outputs and the source image, and compute various metrics which evaluate the model's confidence in the source class on all triangular convex hulls enclosed by the source image and any two target images. We observe that the extent of convexity is greatly pronounced on CIFAR-10, with high confidence maintained over a significant fraction of the convex hull. Indeed, to quantify these observations we use the following metrics: (a) **Average Triangle ($\triangle$) Confidence**: the mean of the model's source class confidence over the enclosed triangle, (b) **Average Triangle ($\triangle$) Fraction** for various values of $\delta$: the fraction of images in the triangular region for which the model confidence is greater than $p_{\text{src}} - \delta$, averaged over all possible target pairs. Here $p_{\text{src}}$ is the confidence of the source image, (c) **Average Path Confidence**: the average model confidence overall linear paths from the source image to all target images. The higher these metrics, the more confident, and thus invariant, the model is in this region. For all of these metrics, we use ten linear interpolations between the source and each of the LST output images and around 65 images that cover the triangular region. We present these metrics in Table 4, along with the mean (and standard deviation of) confidence of the model on the source class ($p_{\text{src}}$) for reference. We thus observe that the average confidence over the triangular convex hull is extremely high on average, and that the super-level sets corresponding to $\delta = 0, 0.1, 0.2, 0.3$ cover a significant fraction of the triangular hull. Further, we observe that the linear interpolant paths maintain a very high degree of confidence as well.

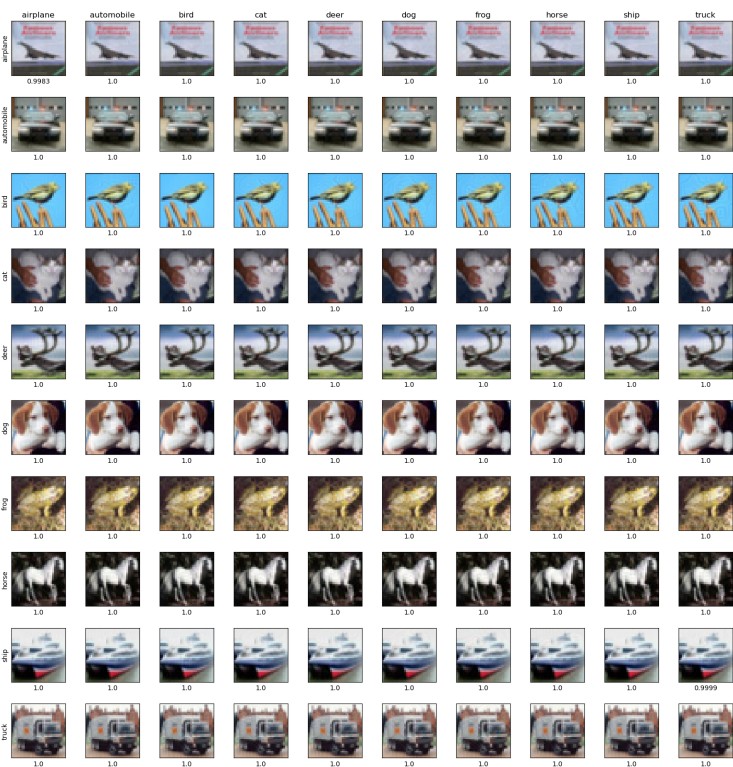

Figure 9: CIFAR10 dataset: Images returned by the Level Set Traversal algorithm for an adversarially trained WideResNet-28 model.

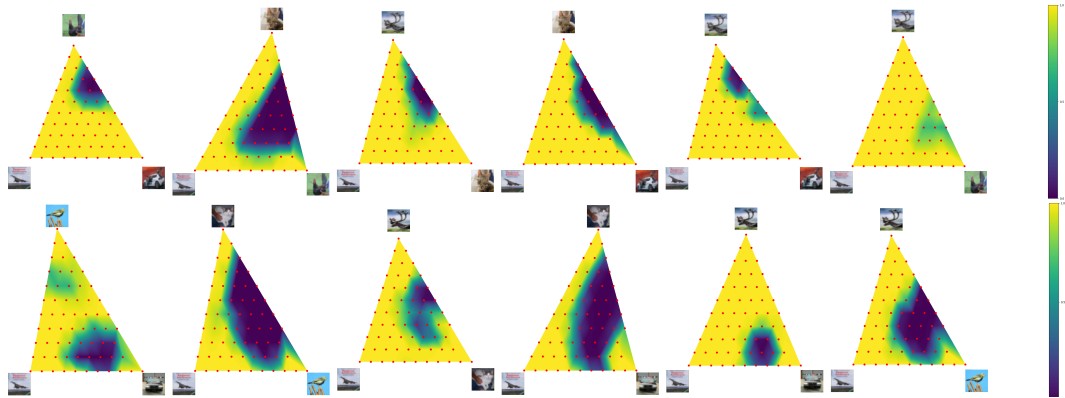

Figure 10: Triangular convex hull enclosed by LST outputs and source images for CIFAR-10 for (top) normally trained WideResNet-28, (bottom) adversarially trained WideResNet-28. In all source-target image pairs, the linear interpolant path maintains high confidence, implying that the source image is linearly connected to the LST target output in a star-like substructure within the level set.

# D    Limitations and Computational Complexity

We note that the proposed Level Set Traversal algorithm typically requires around 200 - 400 iterations to find images which are near-identical to the target image while still maintaining high confidence for the source class since both the semantic and pixel level changes required are of relatively large magnitude. If restricted to a much smaller budget such as 10 steps, we observe that the final image is visually distinct from the target image. Thus, it is non-trivial to design experiments which explicitly train new networks against outputs found by our proposed LST algorithm, akin to that performed

Table 3: Quantitative image distance metrics between output of Level Set Traversal and target images with WideResNet on CIFAR-10.

| Models | RMSE dist: $\mu \pm \sigma$ | $\ell_\infty$ dist: $\mu \pm \sigma$ | SSIM: $\mu \pm \sigma$ | LPIPS dist: $\mu \pm \sigma$ |
|---|---|---|---|---|
| WideResNet (Normal) | $0.020 \pm 0.009$ | $0.071 \pm 0.033$ | $0.985 \pm 0.03$ | $0.001 \pm 0.004$ |
| Trades [Zhang et al., 2019b] | $0.109 \pm 0.044$ | $0.599 \pm 0.123$ | $0.742 \pm 0.132$ | $0.074 \pm 0.037$ |
| Rade and Moosavi-Dezfooli [2022] | $0.113 \pm 0.033$ | $0.652 \pm 0.117$ | $0.717 \pm 0.111$ | $0.079 \pm 0.043$ |
| Wu et al. [2020] | $0.127 \pm 0.032$ | $0.653 \pm 0.12$ | $0.695 \pm 0.105$ | $0.09 \pm 0.042$ |

Table 4: Quantitative measures of model confidence invariance over the triangular convex hull ($\Delta$) of a given source image and all possible target image pairs and over linear interpolant paths between all possible source-target image pairs, using WideResNet on CIFAR-10.

| Models | $p_{\text{src}}$ ($\mu \pm \sigma$) | Avg $\Delta$ Conf. ($\mu \pm \sigma$) | Avg $\Delta$ Frac. ($\mu \pm \sigma$) | | | | Avg Path Conf. ($\mu \pm \sigma$) |
|---|---|---|---|---|---|---|---|
| | | | $\delta = 0.0$ | $\delta = 0.1$ | $\delta = 0.2$ | $\delta = 0.3$ | |
| WideResNet (Normal) | $0.994 \pm 0.033$ | $0.857 \pm 0.097$ | $0.364 \pm 0.228$ | $0.81 \pm 0.113$ | $0.83 \pm 0.109$ | $0.842 \pm 0.105$ | $0.985 \pm 0.031$ |
| Trades [Zhang et al., 2019b] | $0.745 \pm 0.2$ | $0.888 \pm 0.093$ | $0.831 \pm 0.233$ | $0.994 \pm 0.021$ | $0.999 \pm 0.007$ | $0.999 \pm 0.004$ | $0.878 \pm 0.099$ |
| Rade and Moosavi-Dezfooli [2022] | $0.745 \pm 0.18$ | $0.873 \pm 0.06$ | $0.834 \pm 0.212$ | $0.995 \pm 0.02$ | $0.999 \pm 0.005$ | $1.0 \pm 0.002$ | $0.862 \pm 0.073$ |
| Wu et al. [2020] | $0.629 \pm 0.196$ | $0.785 \pm 0.1$ | $0.891 \pm 0.199$ | $0.997 \pm 0.017$ | $1.0 \pm 0.003$ | $1.0 \pm 0.0$ | $0.77 \pm 0.107$ |

within standard adversarial training [Madry et al., 2018]. Furthermore, our quantitative experiments could be potentially extended over higher-dimensional simplices as well, if the curse of dimensionality could be addressed effectively.

### D.1 Best and Worst case scenarios for LST based exploration

Given that the LST algorithm depends primarily on the parallel and orthogonal components of the model gradient, LST is efficacious on deep networks that do not suffer from gradient obfuscation or gradient masking issues, which encompasses nearly all commonly used vision models. On the other hand, for models with non-differentiable or randomized inference-time components, the LST algorithm would have to be modified to utilize techniques such as Backward Pass Differentiable Approximation (BPDA) and Expectation over Transformation (EoT), as proposed in past adversarial attack literature [Athalye et al., 2018], that helps overcome shattered, stochastic, exploding and vanishing gradients. We also note that these modifications also often require a sizable increase in the maximum iteration count to achieve adequate convergence levels.

### D.2 Wall Clock Time

We record wall clock time on a single RTXA5000 GPU with a ResNet-50 model on ImageNet, using a batchsize of 100, and report mean and standard deviation ($\mu \pm \sigma$) statistics over 5 independent minibatches. The proposed LST algorithm is seen to require 101.8 seconds ($\pm$ 0.4s) for 400 iterations, and 52.0 seconds ($\pm$0.3s) for 200 iterations. In comparison, a 100-step PGD attack of the same model takes 25.1 seconds ($\pm$1.4s). Thus, per iteration, the LST algorithm takes the same amount of time as a PGD attack iteration (both about 0.25 seconds). In total, the LST algorithm of 200/400 iterations requires 2x/4x the computation time of a standard 100-step PGD adversarial attack.

We generally observe that for adversarially robust models, near-optimal convergence to the target image (wherein LST output looks essentially identical to the target image) requires a larger number of steps as compared to normally trained models. While we fix the max iterations to 400 across all ImageNet models for instance, it is possible to reduce this parameter for normally trained models while maintaining good visual quality.

## E Ablation and Sensitivity Analysis

Here we perform ablation studies on LST hyperparameters to study the impact of each one independently. We use default parameters — number of LST iterations $m = 400$, step size perpendicular to the gradient $\eta = 10^{-2}$, step size parallel to the gradient $\epsilon = 0.002$, confidence threshold $\delta = 0.2$ — unless mentioned otherwise. We evaluate our ablative analysis on a set of 500 images from ImageNet using a normally trained ResNet-50 model. In general, we observe a trade-off between optimizing for

visual similarity towards the target image of another class, and high-confidence metrics associated with the LST blindspot output so produced.

**A1.** Varying number of iterations $m$ for which LST is run (see Tables 5 and 6): As the number of iterations increases, the distance metrics between the output of LST and the final images decrease. We ideally desire these deviations between LST outputs and target images to be low, to ensure good visual quality. Hence, a higher $m$ value is preferred for performing LST.

**A2.** Varying adversarial step size $\epsilon$ parallel to the negative gradient for LST iterations $m = 100$ (see Tables 7 and 8): With fewer iterations for LST, a larger adversarial step size $\epsilon$ is greatly beneficial. As shown in the Tables, a larger $\epsilon$ results in better confidence metrics over the triangular convex hulls.

**A3.** Varying step size $\eta$ perpendicular to the gradient for LST iterations $m = 100$ (see Tables 9 and 10): From the Tables, we can see that as the step size $\eta$ increases, LST output images become closer to the target images for a smaller value of $m = 100$. Also, with larger $\eta$, we can achieve larger triangular confidence scores. However, the best confidence scores are achieved using larger $m$ values.

**A4.** Varying step size $\eta$ perpendicular to the gradient (see Tables 11 and 12): As expected, the distance metrics between the LST output images and the target images are larger with a lower $\eta$. $\eta = 0.01, 0.05$ achieve desirable small distance scores. Among these, our default parameter $\eta = 0.05$ achieves the best confidence scores, alongside adequate convergence with minimal distance metrics towards the target image.

**A5.** Varying step size $\epsilon$ parallel to the gradient (see Tables 13 and 14): A larger $\epsilon$ results in LST output images that are closer to the final images in terms of our distance metrics. We also note that a larger adversarial perturbation helps in achieving higher triangular confidence scores. We observe that setting the adversarial step size to zero results in output images that are perceptually different compared to the target image, as observed with the larger image distance metrics such as LPIPS distance. However, we also observe that even extremely small non-zero adversarial step-sizes are very effective in improving convergence towards the target images.

Table 5: **A1.** Varying number of iterations $m$ for which LST is run.

| $m$ | RMSE: $\mu \pm \sigma$ | $\ell_\infty$ dist: $\mu \pm \sigma$ | SSIM: $\mu \pm \sigma$ | LPIPS dist: $\mu \pm \sigma$ |
|---|---|---|---|---|
| 100 | $0.141 \pm 0.029$ | $0.351 \pm 0.018$ | $0.715 \pm 0.1$ | $0.288 \pm 0.109$ |
| 200 | $0.052 \pm 0.01$ | $0.133 \pm 0.011$ | $0.913 \pm 0.074$ | $0.064 \pm 0.051$ |
| 300 | $0.019 \pm 0.004$ | $0.063 \pm 0.015$ | $0.971 \pm 0.05$ | $0.011 \pm 0.013$ |
| 400 | $0.008 \pm 0.001$ | $0.045 \pm 0.019$ | $0.989 \pm 0.025$ | $0.002 \pm 0.002$ |

Table 6: **A1.** Varying number of iterations $m$ for which LST is run.

| $m$ | $p_{\text{src}}$ $(\mu \pm \sigma)$ | Avg $\Delta$ Conf. $(\mu \pm \sigma)$ | $\delta = 0.0$ | Avg $\Delta$ Frac. $(\mu \pm \sigma)$ $\delta = 0.1$ | $\delta = 0.2$ | $\delta = 0.3$ | Avg Path Conf. $(\mu \pm \sigma)$ |
|---|---|---|---|---|---|---|---|
| 100 | $0.996 \pm 0.022$ | $0.91 \pm 0.111$ | $0.333 \pm 0.291$ | $0.876 \pm 0.134$ | $0.892 \pm 0.125$ | $0.901 \pm 0.119$ | $0.999 \pm 0.007$ |
| 200 | $0.996 \pm 0.022$ | $0.702 \pm 0.128$ | $0.204 \pm 0.211$ | $0.654 \pm 0.139$ | $0.673 \pm 0.135$ | $0.686 \pm 0.132$ | $0.996 \pm 0.015$ |
| 300 | $0.996 \pm 0.022$ | $0.611 \pm 0.121$ | $0.157 \pm 0.18$ | $0.562 \pm 0.131$ | $0.582 \pm 0.128$ | $0.595 \pm 0.126$ | $0.985 \pm 0.034$ |
| 400 | $0.996 \pm 0.022$ | $0.564 \pm 0.119$ | $0.138 \pm 0.162$ | $0.512 \pm 0.129$ | $0.534 \pm 0.126$ | $0.547 \pm 0.124$ | $0.96 \pm 0.062$ |

Table 7: **A2.** Varying adversarial step size $\epsilon$ for LST iterations $m = 100$.

| $\epsilon$ | RMSE: $\mu \pm \sigma$ | $\ell_\infty$ dist: $\mu \pm \sigma$ | SSIM: $\mu \pm \sigma$ | LPIPS dist: $\mu \pm \sigma$ |
|---|---|---|---|---|
| 0.002 | $0.141 \pm 0.029$ | $0.351 \pm 0.018$ | $0.715 \pm 0.1$ | $0.288 \pm 0.109$ |
| 0.004 | $0.141 \pm 0.029$ | $0.351 \pm 0.018$ | $0.715 \pm 0.1$ | $0.288 \pm 0.109$ |
| 0.008 | $0.141 \pm 0.029$ | $0.351 \pm 0.02$ | $0.715 \pm 0.1$ | $0.288 \pm 0.109$ |
| 0.012 | $0.141 \pm 0.029$ | $0.352 \pm 0.022$ | $0.715 \pm 0.1$ | $0.289 \pm 0.109$ |

Table 8: **A2.** Varying adversarial step size $\epsilon$ for LST iterations $m = 100$.

| $\epsilon$ | $p_{\text{src}}$ ($\mu \pm \sigma$) | Avg $\Delta$ Conf. ($\mu \pm \sigma$) | Avg $\Delta$ Frac. ($\mu \pm \sigma$) $\delta = 0.0$ | $\delta = 0.1$ | $\delta = 0.2$ | $\delta = 0.3$ | Avg Path Conf. ($\mu \pm \sigma$) |
|---|---|---|---|---|---|---|---|
| 0.002 | $0.996 \pm 0.022$ | $0.91 \pm 0.111$ | $0.333 \pm 0.291$ | $0.876 \pm 0.134$ | $0.892 \pm 0.125$ | $0.901 \pm 0.119$ | $0.999 \pm 0.007$ |
| 0.004 | $0.996 \pm 0.022$ | $0.914 \pm 0.107$ | $0.353 \pm 0.3$ | $0.882 \pm 0.129$ | $0.897 \pm 0.121$ | $0.905 \pm 0.116$ | $0.999 \pm 0.006$ |
| 0.008 | $0.996 \pm 0.022$ | $0.918 \pm 0.104$ | $0.377 \pm 0.309$ | $0.887 \pm 0.125$ | $0.901 \pm 0.118$ | $0.909 \pm 0.112$ | $0.999 \pm 0.005$ |
| 0.012 | $0.996 \pm 0.022$ | $0.92 \pm 0.102$ | $0.393 \pm 0.314$ | $0.89 \pm 0.122$ | $0.904 \pm 0.115$ | $0.912 \pm 0.11$ | $0.999 \pm 0.004$ |

Table 9: **A3.** Varying step size $\eta$ for LST iterations $m = 100$.

| $\eta$ | RMSE: $\mu \pm \sigma$ | $\ell_\infty$ dist: $\mu \pm \sigma$ | SSIM: $\mu \pm \sigma$ | LPIPS dist: $\mu \pm \sigma$ |
|---|---|---|---|---|
| 0.01 | $0.141 \pm 0.029$ | $0.351 \pm 0.018$ | $0.715 \pm 0.1$ | $0.288 \pm 0.109$ |
| 0.02 | $0.051 \pm 0.011$ | $0.132 \pm 0.015$ | $0.914 \pm 0.074$ | $0.063 \pm 0.052$ |
| 0.03 | $0.019 \pm 0.006$ | $0.062 \pm 0.02$ | $0.972 \pm 0.05$ | $0.01 \pm 0.019$ |
| 0.04 | $0.008 \pm 0.012$ | $0.046 \pm 0.034$ | $0.988 \pm 0.035$ | $0.004 \pm 0.026$ |

Table 10: **A3.** Varying step size $\eta$ for LST iterations $m = 100$.

| $\eta$ | $p_{\text{src}}$ ($\mu \pm \sigma$) | Avg $\Delta$ Conf. ($\mu \pm \sigma$) | Avg $\Delta$ Frac. ($\mu \pm \sigma$) $\delta = 0.0$ | $\delta = 0.1$ | $\delta = 0.2$ | $\delta = 0.3$ | Avg Path Conf. ($\mu \pm \sigma$) |
|---|---|---|---|---|---|---|---|
| 0.01 | $0.996 \pm 0.022$ | $0.91 \pm 0.111$ | $0.333 \pm 0.291$ | $0.876 \pm 0.134$ | $0.892 \pm 0.125$ | $0.901 \pm 0.119$ | $0.999 \pm 0.007$ |
| 0.02 | $0.996 \pm 0.022$ | $0.704 \pm 0.122$ | $0.18 \pm 0.197$ | $0.655 \pm 0.132$ | $0.676 \pm 0.129$ | $0.688 \pm 0.127$ | $0.997 \pm 0.011$ |
| 0.03 | $0.996 \pm 0.022$ | $0.637 \pm 0.108$ | $0.139 \pm 0.169$ | $0.59 \pm 0.116$ | $0.61 \pm 0.114$ | $0.622 \pm 0.113$ | $0.995 \pm 0.014$ |
| 0.04 | $0.996 \pm 0.022$ | $0.622 \pm 0.102$ | $0.13 \pm 0.16$ | $0.574 \pm 0.108$ | $0.594 \pm 0.108$ | $0.607 \pm 0.106$ | $0.993 \pm 0.017$ |

Table 11: **A4.** Varying step size $\eta$ for LST.

| $\eta$ | RMSE: $\mu \pm \sigma$ | $\ell_\infty$ dist: $\mu \pm \sigma$ | SSIM: $\mu \pm \sigma$ | LPIPS dist: $\mu \pm \sigma$ |
|---|---|---|---|---|
| 0.001 | $0.257 \pm 0.052$ | $0.642 \pm 0.036$ | $0.433 \pm 0.093$ | $0.573 \pm 0.089$ |
| 0.005 | $0.052 \pm 0.01$ | $0.135 \pm 0.015$ | $0.912 \pm 0.074$ | $0.065 \pm 0.051$ |
| 0.01 | $0.008 \pm 0.001$ | $0.045 \pm 0.019$ | $0.989 \pm 0.025$ | $0.002 \pm 0.002$ |
| 0.05 | $0.004 \pm 0.013$ | $0.04 \pm 0.032$ | $0.994 \pm 0.027$ | $0.003 \pm 0.024$ |

Table 12: **A4.** Varying step size $\eta$ for LST.

| $\eta$ | $p_{\text{src}}$ ($\mu \pm \sigma$) | Avg $\Delta$ Conf. ($\mu \pm \sigma$) | Avg $\Delta$ Frac. ($\mu \pm \sigma$) $\delta = 0.0$ | $\delta = 0.1$ | $\delta = 0.2$ | $\delta = 0.3$ | Avg Path Conf. ($\mu \pm \sigma$) |
|---|---|---|---|---|---|---|---|
| 0.001 | $0.996 \pm 0.022$ | $0.998 \pm 0.021$ | $0.642 \pm 0.379$ | $0.997 \pm 0.03$ | $0.998 \pm 0.026$ | $0.998 \pm 0.024$ | $1.0 \pm 0.002$ |
| 0.005 | $0.996 \pm 0.022$ | $0.707 \pm 0.129$ | $0.237 \pm 0.227$ | $0.661 \pm 0.14$ | $0.679 \pm 0.136$ | $0.69 \pm 0.134$ | $0.996 \pm 0.017$ |
| 0.01 | $0.996 \pm 0.022$ | $0.564 \pm 0.119$ | $0.138 \pm 0.162$ | $0.512 \pm 0.129$ | $0.534 \pm 0.126$ | $0.547 \pm 0.124$ | $0.96 \pm 0.062$ |
| 0.05 | $0.996 \pm 0.022$ | $0.61 \pm 0.1$ | $0.131 \pm 0.154$ | $0.561 \pm 0.108$ | $0.582 \pm 0.104$ | $0.595 \pm 0.103$ | $0.985 \pm 0.03$ |

Table 13: **A5.** Varying adversarial stepsize $\epsilon$ for which LST is run.

| $\epsilon$ | RMSE: $\mu \pm \sigma$ | $\ell_\infty$ dist: $\mu \pm \sigma$ | SSIM: $\mu \pm \sigma$ | LPIPS dist: $\mu \pm \sigma$ |
|---|---|---|---|---|
| 0.02 | $0.008 \pm 0.003$ | $0.055 \pm 0.055$ | $0.986 \pm 0.03$ | $0.005 \pm 0.015$ |
| 0.002 | $0.008 \pm 0.001$ | $0.045 \pm 0.019$ | $0.989 \pm 0.025$ | $0.002 \pm 0.002$ |
| 0.0002 | $0.007 \pm 0.001$ | $0.04 \pm 0.013$ | $0.99 \pm 0.025$ | $0.002 \pm 0.002$ |
| 0.0 | $0.093 \pm 0.063$ | $0.234 \pm 0.147$ | $0.83 \pm 0.129$ | $0.147 \pm 0.118$ |

Table 14: **A5.** Varying adversarial stepsize $\epsilon$ for which LST is run.

| $\epsilon$ | $p_{\text{src}}$ ($\mu \pm \sigma$) | Avg $\Delta$ Conf. ($\mu \pm \sigma$) | Avg $\Delta$ Frac. ($\mu \pm \sigma$) $\delta = 0.0$ | $\delta = 0.1$ | $\delta = 0.2$ | $\delta = 0.3$ | Avg Path Conf. ($\mu \pm \sigma$) |
|---|---|---|---|---|---|---|---|
| 0.02 | $0.996 \pm 0.022$ | $0.596 \pm 0.114$ | $0.193 \pm 0.195$ | $0.551 \pm 0.122$ | $0.57 \pm 0.119$ | $0.581 \pm 0.119$ | $0.981 \pm 0.042$ |
| 0.002 | $0.996 \pm 0.022$ | $0.564 \pm 0.119$ | $0.138 \pm 0.162$ | $0.512 \pm 0.129$ | $0.534 \pm 0.126$ | $0.547 \pm 0.124$ | $0.96 \pm 0.062$ |
| 0.0002 | $0.996 \pm 0.022$ | $0.527 \pm 0.125$ | $0.097 \pm 0.129$ | $0.464 \pm 0.135$ | $0.491 \pm 0.132$ | $0.507 \pm 0.131$ | $0.92 \pm 0.088$ |
| 0.0 | $0.996 \pm 0.022$ | $0.741 \pm 0.104$ | $0.148 \pm 0.177$ | $0.656 \pm 0.126$ | $0.715 \pm 0.12$ | $0.731 \pm 0.114$ | $0.97 \pm 0.032$ |

# F    Additional Level Set Traversal Examples

Figures 11, 12, 13, 14, and 15 show examples images output by our LST algorithm using ImageNet images for a normally trained ResNet-50 from ablation studies A1, A2, A3, A4, and A5, respectively. Figures 16, 17, 18, 19, and 20 show their respective triangular convex hull.

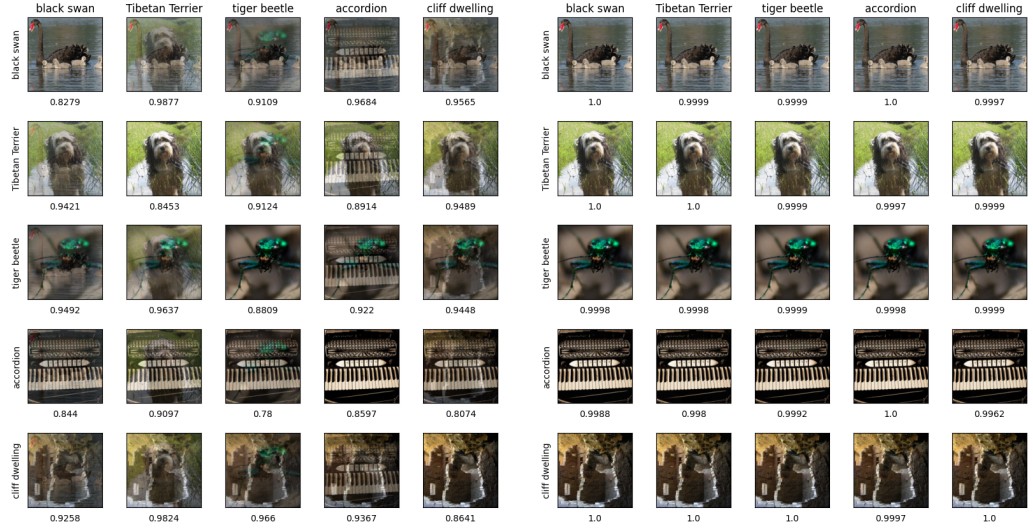

Figure 11: ImageNet images returned by the LST algorithm for a normally trained ResNet-50. These are examples from ablation **A1**. Left: $m = 100$ and Right: $m = 400$.

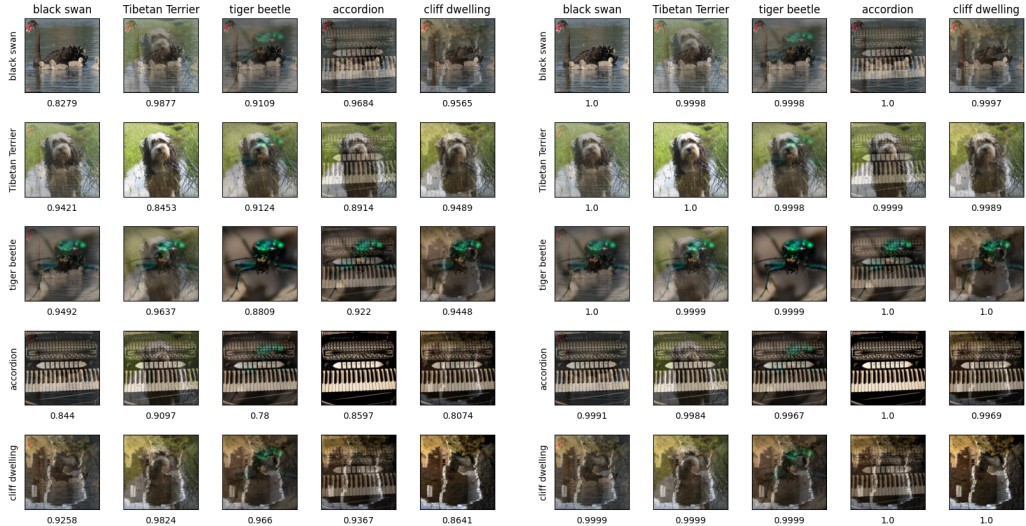

Figure 12: ImageNet images returned by the LST algorithm for a normally trained ResNet-50. These are examples from ablation **A2**. Left: $m = 100$, $\epsilon = 0.002$ and Right: $m = 400$, $\epsilon = 0.012$.

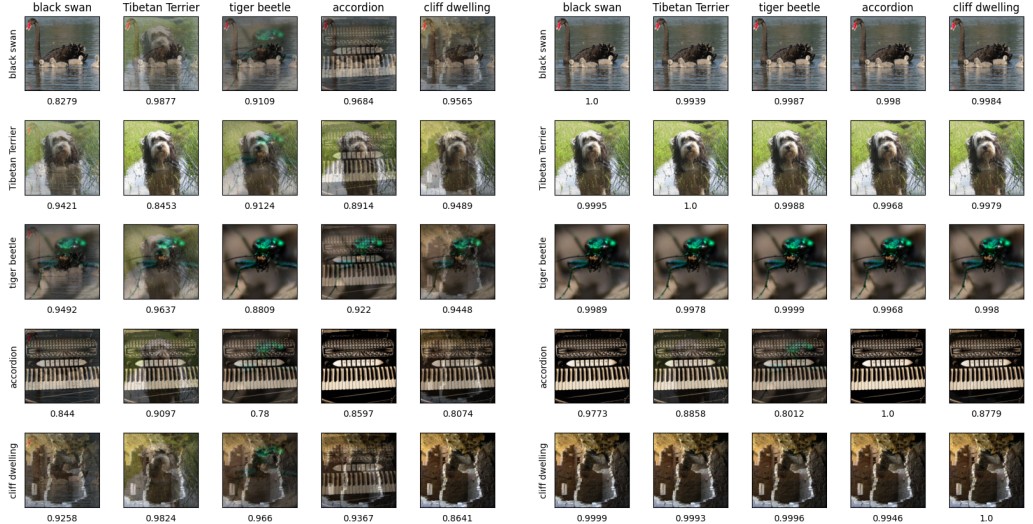

Figure 13: ImageNet images returned by the LST algorithm for a normally trained ResNet-50. These are examples from ablation **A3**. Left: $m = 100$, $\eta = 0.01$ and Right: $m = 400$, $\eta = 0.04$.

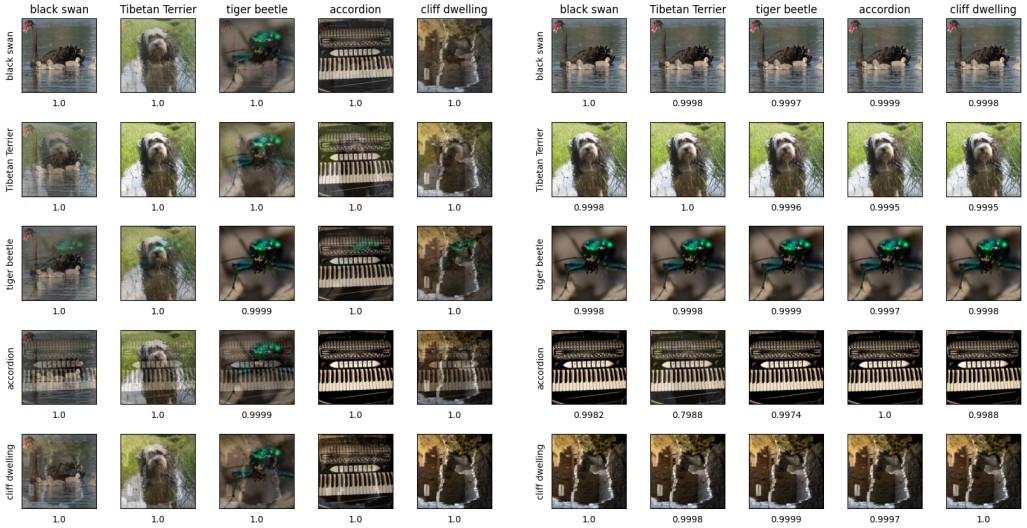

Figure 14: ImageNet images returned by the LST algorithm for a normally trained ResNet-50. These are examples from ablation **A4**. Left: $\eta = 0.001$ and Right: $\eta = 0.05$.

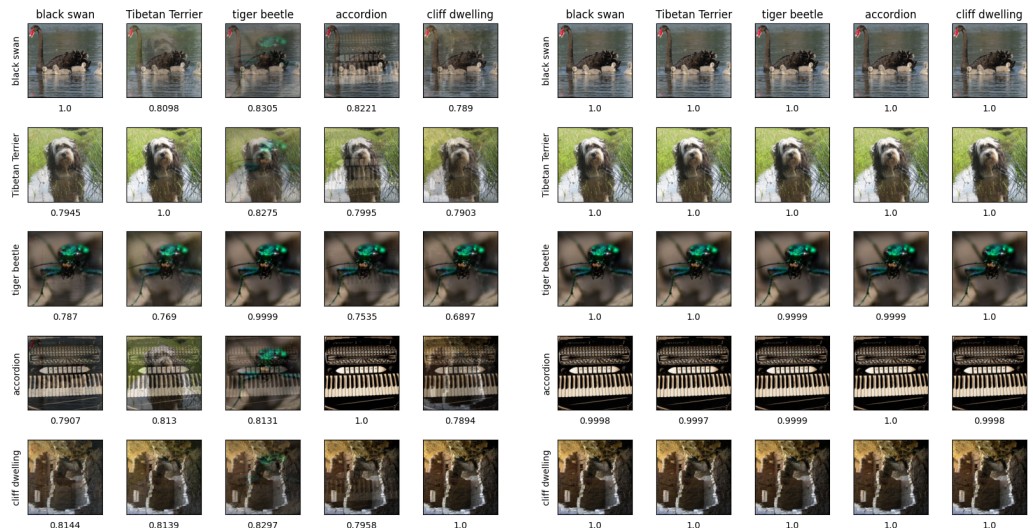

Figure 15: ImageNet images returned by the LST algorithm for a normally trained ResNet-50. These are examples from ablation **A5**. Left: $\epsilon = 0.0$ and Right: $\epsilon = 0.02$.

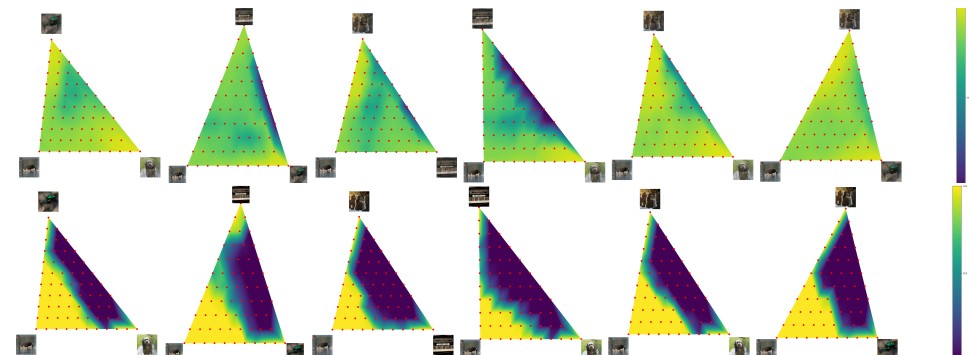

Figure 16: Triangular convex hull enclosed by LST outputs and source images for ImageNet with normally trained ResNet-50. These are examples from ablation **A1**. Top: $m = 100$ and Bottom: $m = 400$.

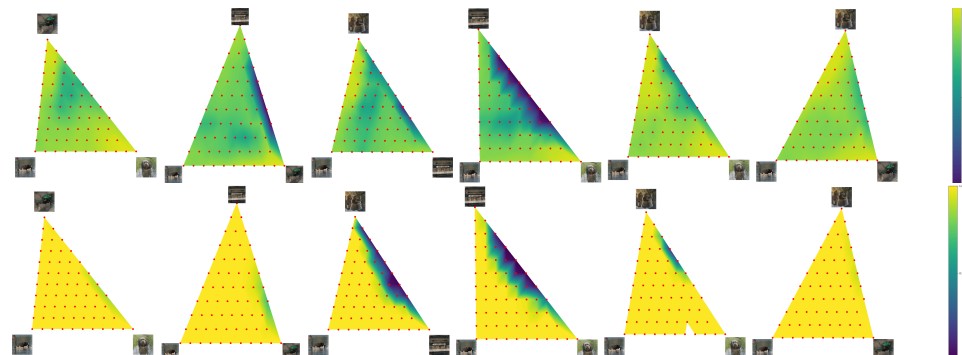

Figure 17: Triangular convex hull enclosed by LST outputs and source images for ImageNet with normally trained ResNet-50. These are examples from ablation **A2**. Top: $m = 100$, $\epsilon = 0.002$ and Bottom: $m = 400$, $\epsilon = 0.012$. In the latter setting, the confidences are extremely high given that the final images as indicated in Fig.-12 are not extremely similar to the final target images.

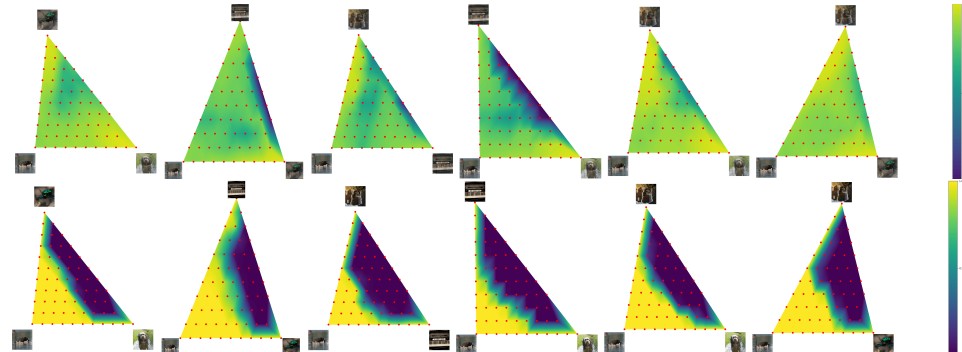

Figure 18: Triangular convex hull enclosed by LST outputs and source images for ImageNet with normally trained ResNet-50. These are examples from ablation **A3**. Top: $m = 100$, $\eta = 0.01$ and Bottom: $m = 400$, $\eta = 0.04$.

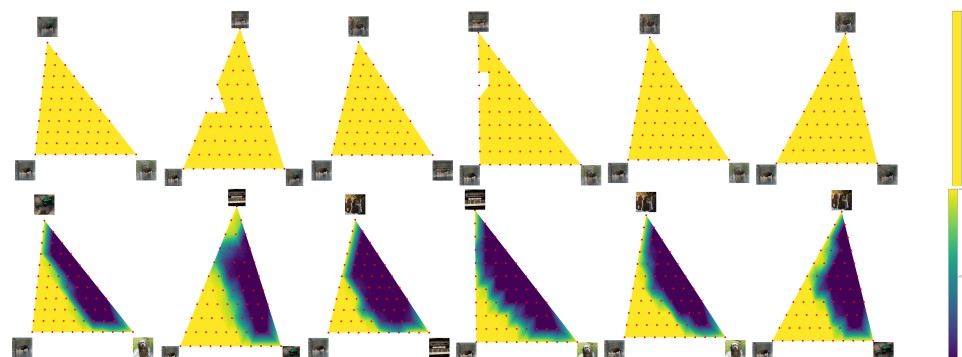

Figure 19: Triangular convex hull enclosed by LST outputs and source images for ImageNet with normally trained ResNet-50. These are examples from ablation **A4**. Top: $\eta = 0.001$ and Bottom: $\eta = 0.05$. In this setting with $\eta = 0.001$, the confidences are extremely high given that the final images as indicated in Fig.-14 are not sufficiently close to the final target images.

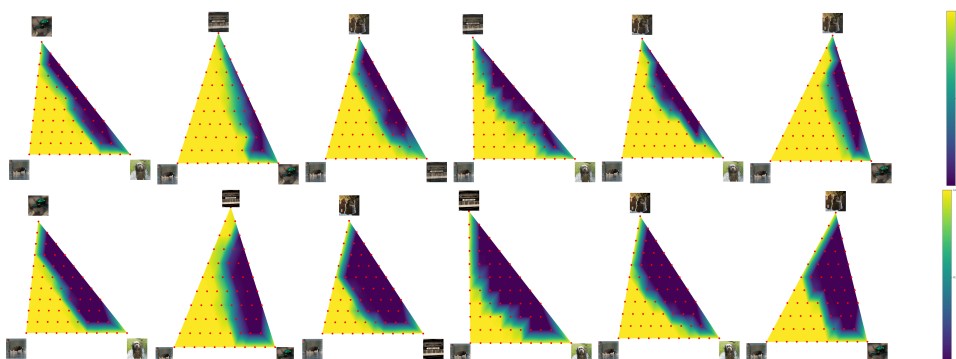

Figure 20: Triangular convex hull enclosed by LST outputs and source images for ImageNet with normally trained ResNet-50. These are examples from ablation **A5**. Top: $\epsilon = 0.0$ and Bottom: $\epsilon = 0.02$.

