# OpenReview forum: "Exploring Geometry of Blind Spots in Vision models"
_NeurIPS.cc/2023/Conference — NeurIPS 2023 spotlight_

### Official Review · Reviewer_yXeg · 2023-06-29

**Soundness:** 4 excellent
**Presentation:** 3 good
**Contribution:** 3 good
**Rating:** 7
**Confidence:** 3

**Summary:**

The authors explore the output space of vision models. Specifically, they propose an algorithm (Level Set Traversal, LST) which, starting from a "source" input image transforms it into an image that looks completely different (e.g. an image from a different class) while still confidently predicting the "source" class. This proves that neural networks have "blind spots" they further analyze the behavior of networks, and show that the paths between these inputs are connected.

**Strengths:**

Originality: the paper improves upon previous work in the area, which only applied to very specific network architectures (invertible resnets). In contrast, the novel LST algorithm can be applied to virtually any network.

Quality: Both the theoretical and the empirical work appear solid to me.

Clarity: I was not able to follow all the math in detail, but was nonetheless able to follow along nicely.

Significance: I think this paper is a solid contribution to the field. It provides both some theoretical analysis of the output space of vision models, as well as an algorithm for producing "blind spot" examples that is easy to understand and implement, and seems to work well.

**Weaknesses:**

* The authors state that the algorithm takes 200-400 iterations for each input image. To put this into context it would be nice if the authors could give a wall clock time estimate on how long this would take.


**Questions:**

* The authors mention that doing only 10ish iterations produces much lower-quality results. I would be curious to see what these would look like (though this is just personal curiosity, so feel free to ignore this).

* I don't understand what the colors within the triangles in Figure 4 represents, could you explain it?

* Do the images you obtain transfer to other architectures? E.g. If we take the off-diagonal images from Fig. 2 and put them into a different network than the one used to create them, would they be classified as target or source class?

**Limitations:**

The authors have adequately discusses limitations.

---

> ### Author Rebuttal · Authors · 2023-08-10
>
> We thank the reviewer for their valuable feedback. We are encouraged that the reviewer found that the proposed method is original, explained clearly, and is a significant contribution to the field. We respond to the questions raised below:
>
> > The authors state that the algorithm takes 200-400 iterations for each input image. To put this into context it would be nice if the authors could give a wall clock time estimate on how long this would take.
> - *Wall Clock Time:* We record wall clock time on a single RTXA5000 GPU with a ResNet-50 model on ImageNet, using a batchsize of 100, and report mean and standard deviation ($\mu \pm \sigma$) statistics over 5 independent minibatches.
> The proposed LST algorithm is seen to require 101.8 seconds ($\pm$ 0.4s) for 400 iterations, and 52.0 seconds ($\pm$0.3s) for 200 iterations.
> In comparison, a 100-step PGD attack of the same model takes 25.1 seconds ($\pm$1.4s). Thus, per iteration, the LST algorithm takes the same amount of time as a PGD attack iteration (both about 0.25 seconds). In total, the LST algorithm of 200/400 iterations requires 2x/4x the computation time of a standard 100-step PGD adversarial attack.
>
> - In Figure-2 of the Rebuttal pdf, we present the LST outputs as obtained with just 10 iterations. Here, we clearly observe that the LST output images (off-diagonal entries) are not adequately similar perceptually to the target image of interest, but are often rather more similar to the source image. Thus, in practice, we set the number of iterations to be between 200 and 400 to produce LST outputs very similar to the target image.
>
> > I don't understand what the colors within the triangles in Figure 4 represents, could you explain it?
>
> - In order to visualize the extent of level sets over a two-dimensional subset, we evaluate the model confidence over the triangular convex hull obtained by linearly interpolating over three reference points, namely the source image and the two target blindspot images. The prediction confidence (in the range [0,1]) assigned by the model with respect to the source class is mapped to the continuous color-bar (shown in the rightmost part of Figure 4), with high-confidence (close to 1.0) points appearing as bright yellow, and low-confidence (close to 0.0) points appearing as dark violet.
>
>
> > Do the images you obtain transfer to other architectures? E.g. If we take the off-diagonal images from Fig. 2 and put them into a different network than the one used to create them, would they be classified as target or source class?
> - *Transferability of LST outputs:* In general, we observe little transferability between different networks, that is, the off-diagonal images generated by LST on one network are generally assigned low confidence by a different network with respect to the source class. This is likely because the LST algorithm optimizes the final output to be highly specific to the original network (especially with the choice of relatively small step sizes over several iterations), and transferability is not explicitly incorporated into its design. However, we hypothesize that transferability of LST outputs can be improved significantly by incorporating techniques proposed in a vast body of existing literature that works towards boosting transferability of standard adversarial attacks [1,2,3] and Universal Adversarial Perturbations (UAP) [4,5,6].
>
>
> We thank the reviewer for the support for acceptance. We greatly appreciate the valuable comments and suggestions, and we will certainly incorporate them in the final version of the paper.
>
>
> References:
>
> [1] Xie et al.,  Improving Transferability of Adversarial Examples with Input Diversity, CVPR 2019
>
> [2] Wu et al., Boosting the Transferability of Adversarial Samples via Attention, CVPR 2020
>
> [3] Qin et al., Boosting the Transferability of Adversarial Attacks with Reverse Adversarial Perturbation, NeurIPS 2022
>
> [4] Dezfooli et al., Universal adversarial perturbations, CVPR 2017
>
> [5] Mopuri et al., Fast feature fool: A data independent approach to universal adversarial perturbations, BMVC 2017
>
> [6] Benz et al., Double Targeted Universal Adversarial Perturbations, ACCV 2020

---

> > ### Comment · Reviewer_yXeg · 2023-08-15
> >
> > Thank you for the clarifications. I stand by my original review that this manuscript should be accepted.

---

> > > ### Author Response · Authors · 2023-08-18
> > > **Thank you**
> > >
> > > Thank you very much for your response. Again, we are glad that you support the acceptance of our paper.

---

### Official Review · Reviewer_o1gw · 2023-07-07

**Soundness:** 4 excellent
**Presentation:** 4 excellent
**Contribution:** 4 excellent
**Rating:** 7
**Confidence:** 3

**Summary:**

The paper studies the under-sensitivity of such deep neural networks, where it is possible to find large perturbations in the input space (such as transforming one image to another) without significant changes to the activations/predictions. Towards this goal, the paper proposes Level Set Traversal, LST, which adds perturbations in an incremental manner to move along regions orthogonal to the gradient (w.r.t input) space computed on the classification loss. Evaluations are done on ImageNet and CIFAR-10, and results show that LST is able to find not only large perturbations that continue to yield high confidence on the source class, but a path of high confidence from the source to the target. Experiments also show that the convex hull formed by a source image and a pair of LST output images for two targets enclose a region of high-confidence for adversarially trained models, unlike non-adversarially trained ones.

**Strengths:**

- The explanation and intuition of the method is very clear, and it also appears to be highly effective in finding an entire path of high confidence/similar loss from two highly visually different images.
- Theoretical analysis presented in Sec 4 is also helpful and insightful.
- Additional experiments on CIFAR-10 and ablation studies in the supplementary material are also appreciated.

**Weaknesses:**

1) The experiment details mention 1000 images from ImageNet are chosen as the source images, but only 5 target images (the ones in the figure) are used. It is difficult to tell whether the results are overfitted to the 5 target images. More extensive evaluation can be done, for example simply randomly choosing source-target pairs from the 1000 chosen images.

2) I do not see any metric measuring the confidence of the output of LST alone. This metric is highly important especially for Table 1, to show that LST can both jointly optimize for minimal distance from target image as well as preserving confidence. Measuring path confidence averages the results across the entire path is not sufficient since the confidence result can be biased by points closer to the source image and hence more likely to exhibit high confidence.

3) Measurement of computational cost (wall-clock time) of the LST algorithm might be useful.

**Questions:**

What are the main failure cases of LST - under what situations does LST work best, and when is LST unable to find such paths or outputs of low loss/high confidence?

**Limitations:**

Limitations are discussed.

---

> ### Author Rebuttal · Authors · 2023-08-10
>
> We thank the reviewer for their valuable feedback. We are encouraged that the reviewer found that the proposed method is explained clearly and intuitively, is effective in practice and supported by theoretical insights. We respond to the questions raised below:
>
>
> >The experiment details mention 1000 images from ImageNet are chosen as the source images, but only 5 target images (the ones in the figure) are used. It is difficult to tell whether the results are overfitted to the 5 target images. More extensive evaluation can be done, for example simply randomly choosing source-target pairs from the 1000 chosen images.
>
> - *Choice of Target Classes:* For CIFAR-10, we consider all remaining 9 classes as target classes in our evaluations. For ImageNet, we pick 5 arbitrary classes for our experiments in the paper, but observe that the choice of the target images/classes does not affect our results much. To show this, we rerun our experiments twice more for a normally trained ResNet-50 model wherein we randomly sample another set of 5 classes each time. We also randomly sample two target images for each source image in the last row. We present our results in the tables below:
>
>
> | | $\ell_2$ dist: $\mu \pm \sigma$ | $\ell_\infty$ dist: $\mu \pm \sigma$ | SSIM: $\mu \pm \sigma$ | LPIPS dist: $\mu \pm \sigma$ |
> |-------|------------------------ |------------------------ |------------------------ |------------------------ |
> |Paper| 2.6 $\pm$ 1.1       | 0.033 $\pm$ 0.011 | 0.995 $\pm$ 0.006 | 0.003 $\pm$ 0.004|
> |Random sampling 1| 2.8 $\pm$ 0.9 | 0.031 $\pm$ 0.012 | 0.996 $\pm$ 0.004 | 0.002 $\pm$ 0.004|
> |Random sampling 2| 2.9 $\pm$ 1.3 | 0.055 $\pm$ 0.031 | 0.996 $\pm$ 0.005 | 0.002 $\pm$ 0.003|
> | Random sampling per image| 3.5 $\pm$ 4.9 | 0.044 $\pm$ 0.035 | 0.988 $\pm$ 0.029 | 0.003 $\pm$ 0.016|
>
>
>
> | | $~~~~~~p_{\text{src}}$  | Avg $\Delta$ Conf.  | Avg $\Delta$ Frac  $(\delta=0.0)$ | $~~~(\delta=0.1)$ |  $~~(\delta=0.2)$ | $~~(\delta=0.3)$|  Avg Path Conf. |
> |----|------------------------ |------------------------|------------------------ |------------------------|------------------------ |------------------------|------------------------ |
> |Paper| 0.96 $\pm$ 0.12 | 0.49 $\pm$ 0.11 | $~~~~~$0.13 $\pm$ 0.12 | 0.42 $\pm$ 0.12 | 0.46 $\pm$ 0.12| 0.48 $\pm$ 0.11 | 0.87 $\pm$ 0.09 |
> |Random sampling 1| 0.95 $\pm$ 0.14 | 0.47 $\pm$ 0.12 | $~~~~~$0.13 $\pm$ 0.13 | 0.41 $\pm$ 0.13 | 0.44 $\pm$ 0.13 | 0.46 $\pm$ 0.13 | 0.81 $\pm$ 0.11|
> |Random sampling 2| 0.95 $\pm$ 0.13 | 0.42 $\pm$ 0.11 | $~~~~~$0.11 $\pm$ 0.12 | 0.36 $\pm$ 0.11 | 0.39 $\pm$ 0.12 | 0.41 $\pm$ 0.12 | 0.81 $\pm$ 0.11 |
> | Random sampling per image |  0.94 $\pm$ 0.14 | 0.43 $\pm$ 0.14 | $~~~~~$0.11 $\pm$ 0.13 | 0.36 $\pm$ 0.15 | 0.39 $\pm$ 0.15 | 0.42 $\pm$ 0.16 | 0.79 $\pm$ 0.14|
>
>
> -  The confidence of the LST output itself generally depends on a combination of hyperparameter choices, such as the confidence threshold $\delta$, number of steps $m$, and steps sizes for the perpendicular and parallel components to the local gradient. For example, if the step size perpendicular to the gradient is made larger, the network confidence can drop below $p_{src} - \delta$, terminating the LST algorithm. Thus when the LST algorithm stops before the max-iterations is completed, the penultimate point that has network confidence above $p_{src} - \delta$ is set as the LST output. However, early termination is rarely observed with an appropriate choice of hyperparameters. Thus in all cases, the LST output is guaranteed to have network confidence above $p_{src} - \delta$, for all LST image metrics shown in Table-1. For reference, we set the confidence threshold  $\delta=$ 0.2 for ImageNet, which means that the confidence on the LST output never drops below the confidence of the source image by more than 0.2. This can also be seen individually for all LST output images which are the off-diagonal images in Figures-2,3 (original source images are the diagonal images), indicating that the LST outputs have high confidence, generally  well-above the lower bound given by $p_{src} - \delta$ = $p_{src}- $  0.2.
>
> * *Wall Clock Time:* We record wall clock time on a single RTXA5000 GPU with a ResNet-50 model on ImageNet, using a batch size of 100, and report mean and standard deviation ($\mu \pm \sigma$) statistics over 5 independent mini-batches.
> The proposed LST algorithm is seen to require 101.8 seconds ($\pm$ 0.4s) for 400 iterations, and 52.0 seconds ($\pm$0.3s) for 200 iterations.
> In comparison, a 100-step PGD attack of the same model takes 25.1 seconds ($\pm$1.4s). Thus, per iteration, the LST algorithm takes the same amount of time as a PGD attack iteration (both about 0.25 seconds). In total, the LST algorithm of 200/400 iterations requires 2x/4x the computation time of a standard 100-step PGD adversarial attack.
>
> - *Best and Worst cases:* Given that the LST algorithm depends primarily on the parallel and orthogonal components of the model gradient, LST works best on deep networks that do not suffer from gradient obfuscation or gradient masking issues, which encompasses nearly all commonly used vision models. On the other hand, for models with non-differentiable or randomized inference-time components, the LST algorithm would have to be modified to utilize tricks such as Backward Pass Differentiable Approximation (BPDA) and Expectation over Transformation (EoT) tricks as proposed in past adversarial attack literature [Athalye et al. 2018] that helps overcome shattered, stochastic, exploding and vanishing gradients. We also note that these modifications also often require an sizable increase in the maximum iteration count to achieve adequate convergence levels.
>
>
> We thank the reviewer for the support for acceptance and greatly appreciate the valuable comments and suggestions. We kindly ask if the reviewer would consider increasing their score if their concerns or questions have been addressed. We would be glad to engage further during the discussion period.

---

> > ### Comment · Reviewer_o1gw · 2023-08-20
> > **Thank you**
> >
> > The authors' rebuttal has convincingly demonstrated that the results of the experiments are generalizable to more ImageNet classes apart from the initially chosen 5, and that LST, by construction of the algorithm, maintains high confidence across the entire path. I also appreciate the author's additional results on wall-clock timing and discussion of failure cases, which I believe will be nice to incorporate into the revision to paint a more complete picture. Since my concerns have been addressed, I've raised my score to 7.

---

> > > ### Author Response · Authors · 2023-08-20
> > > **Thank you**
> > >
> > > We are glad that the reviewer found our rebuttal convincing and helpful. We sincerely thank the reviewer for the valuable suggestions and detailed feedback, we will certainly incorporate these and associated additional results into the final version of the paper. We thank the reviewer for raising their score, and for supporting acceptance of the paper.

---

### Official Review · Reviewer_N6FN · 2023-07-10

**Soundness:** 4 excellent
**Presentation:** 3 good
**Contribution:** 4 excellent
**Rating:** 8
**Confidence:** 3

**Summary:**

The authors propose a new adversarial attack on the discriminative vision models called Level Set Traversal (LST). Contrary to the previous attacks, this new algorithm exploits the orthogonal component of the network's gradient to produce samples that can bypass existing adversarially-trained classification networks.

Moreover, the authors showcase that the samples obtained using this algorithm lie on the star-shaped manifold of high-confidence predictions. This is new compared to the previous adversarial attacks, which typically yield samples not connected via high-confidence linear paths.

**Strengths:**

+ The presentation of the paper is great.

+ The motivation behind the proposed method is clear. I also like that the paper is theoretically driven and includes experimentally confirmed results on real-world datasets.

+ To me, the results look really convincing and exciting. I was surprised at the existence of the linearly interconnected sets of adversarial examples, even in adversarially trained models.

+ The ablation study of the method is quite extensive and includes robustness tests for multiple hyperparameters.

**Weaknesses:**

- I would suggest the authors try and make connections to the area of model ensembling, where works such as [1] showcased the existence of piecewise-linear paths with low training error in the space of the model weights.

- I also suggest including section 1 of the appendix in the main paper, as well as extending Figure 4 to include the visualization for multiple attack types. Without reading the appendix, the advantage of the proposed algorithm over previous methods remains unclear. Extended proof of Lemma 1, in my opinion, could be instead moved to the appendix, as it breaks the flow of the text.

- The authors have only validated their approach against one type of adversarial defense for each proposed network. Extending such evaluation would benefit the community and provide a benchmark for future works.

[1] Loss Surfaces, Mode Connectivity, and Fast Ensembling of DNNs, NeurIPS 2018

**Questions:**

* Did the authors benchmark multiple state-of-the-art adversarial defense methods against their attack?

* Why does the discovered high-confidence manifold have more high-confidence regions for adversarially-trained models than non-AT variants? Have the authors evaluated multiple AT methods to explore whether or not the shape of this region depends on the network architecture (ResNet vs ViT) or the specific AT method?

**Limitations:**

The limitations were adequately addressed.

---

> ### Author Rebuttal · Authors · 2023-08-10
>
> We thank the reviewer for their valuable feedback. We are encouraged that the reviewer found the paper well-motivated, clear, convincing and exciting. We respond to the questions raised below:
>
> - *Connections to model ensembling:* We thank the reviewer for the suggestion. The existence of piece-wise linear paths in model parameter space that achieve low loss (on a fixed training or testing set of images) as found by several recent works suggest that those model weights lie within the low-loss level sets and imply similar connected topologies of these sets. By altering the fixed choice of training/testing images using some well-defined perturbations/modifications, the loss function itself would change and induce different level sets in model-weight space for each such modification. We hypothesize that it might be possible to jointly consider the model parameter space and input image space as orthogonal components in a combined high-dimensional product space, wherein a meta-level linear functional can be defined to define level sets and connectivity. This potentially could help identify subsets of model weights that generalize well across different variations of the input image distribution.
>
> - We thank the reviewer for feedback on the section organization, we will certainly incorporate it in future versions.
> - *Extending Evaluations and Benchmarking:* We thank the reviewer for the suggestion to create a common benchmark to evaluate the existing models, especially amongst different adversarial defenses. Our codebase is compatible with loading adversarially trained models from the Model-Zoo offered by RobustBench [Croce et al. 2021]. We additionally present results obtained on other popular adversarial defenses such as TRADES [Zhang et al. 2019] and the current best-performing WideResNet-34-10 model on CIFAR-10 listed on RobustBench (trained with extra data by Rade et al. 2021) below:
>
> Image Metrics between LST output and Target Image:
>
> | | $\ell_2$ dist: $\mu \pm \sigma$ | $\ell_\infty$ dist: $\mu \pm \sigma$ | SSIM: $\mu \pm \sigma$ | LPIPS dist: $\mu \pm \sigma$ |
> |-------|------------------------ |------------------------ |------------------------ |------------------------ |
> Trades | 3.6 $\pm$ 2.2 | 0.467 $\pm$ 0.137 | 0.874 $\pm$ 0.114 | 0.033 $\pm$ 0.03|
> Rade et al. | 4.0 $\pm$ 2.6 | 0.51 $\pm$ 0.145 | 0.846 $\pm$ 0.132 | 0.037 $\pm$ 0.034|
>
>
> Quantitative measures ($\mu \pm \sigma$) of model confidence invariance over the triangular convex hull ($\Delta$) of a given source image and all possible target image pairs and over linear interpolant paths between all possible source-target image pairs:
>
> || $~~~~~~p_{\text{src}}$  | Avg $\Delta$ Conf.  | Avg $\Delta$ Frac $(\delta=0.0)~~~$ |  $~~~~(\delta=0.1)$ |  $~~~(\delta=0.2)$ | $~~~(\delta=0.3)$|  Avg Path Conf. |
> |--------------|------------------------ |------------------------|------------------------ |------------------------|------------------------ |------------------------|------------------------ |
> Trades | 0.75 $\pm$ 0.21 | 0.69 $\pm$ 0.21 | $~~~~$0.24 $\pm$ 0.20 | 0.72 $\pm$ 0.21 | 0.87 $\pm$ 0.14 | 0.94 $\pm$ 0.09 | 0.74 $\pm$ 0.20|
> Rade et al. | 0.75 $\pm$ 0.18 | 0.70 $\pm$ 0.19 | $~~~~$0.24 $\pm$ 0.18 | 0.77 $\pm$ 0.19 | 0.90 $\pm$ 0.11 | 0.96 $\pm$ 0.07 | 0.74 $\pm$ 0.17|
> ---
>
>
> >The authors have only validated their approach against one type of adversarial defense for each proposed network. Extending such evaluation would benefit the community and provide a benchmark for future works.
> >  Did the authors benchmark multiple state-of-the-art adversarial defense methods against their attack?
> > Have the authors evaluated multiple AT methods to explore whether or not the shape of this region depends on the network architecture (ResNet vs ViT) or the specific AT method?
>
> - *Shape of Level Sets:* Due to differences in architecture between ResNet and ViT networks, and differences in training methodologies, the level sets do tend to have different shapes in the two cases. For instance, the ResNet-50 AT model is more confident over a more expansive set of inputs as compared to the DeiT-S AT model, indicated by the larger fraction of the triangular convex hull of a given source image and all possible target image pairs for various confidence threshold settings as presented in Table-2. Between different adversarial training methods on the same architecture, we observe that the extent of the level set generally depends on the extent of regularization utilized, as it explicitly promotes smoothness of network outputs over a predefined input region, and that the AT models tend to be under-sensitive over subsets of the input domain that lie well beyond its original threat model.
>
> > Why does the discovered high-confidence manifold have more high-confidence regions for adversarially-trained models than non-AT variants?
>
> - During adversarial training, the model is explicitly regularized to mitigate oversensitivity to perturbations in input space within the subset of images as defined by the original threat model of interest. It is well known that adversarially trained models are smoother than normally trained models within the original threat model, given that large fluctuations in model prediction are discouraged during adversarial training. We suspect that these adversarially trained models also tend to be smoother over subsets of the input domain that lie well beyond its original threat model due to the regularization encountered during adversarial training, and thus the high confidence region is also more expansive as compared to normally trained models.
>
>
>
> We thank the reviewer for the strong support for acceptance. We greatly appreciate the valuable comments and suggestions, and we will certainly incorporate them in the final version of the paper.

---

> > ### Comment · Reviewer_N6FN · 2023-08-17
> > **Great rebuttal, I keep my rating**
> >
> > I would like to thank the authors for the comprehensive rebuttal and keep my initial rating. In case of acceptance, I recommend the authors incorporate the additional presented experiments and rework the text for better clarity and comprehensiveness in the camera-ready version.

---

> > > ### Author Response · Authors · 2023-08-18
> > > **Thank you**
> > >
> > > We are glad that you found our rebuttal comprehensive and helpful. We will certainly incorporate the additional results and rephrase the text in the camera-ready version as suggested. Once again, we thank you for your strong support for acceptance of the paper.

---

### Official Review · Reviewer_LWdq · 2023-07-11

**Soundness:** 3 good
**Presentation:** 3 good
**Contribution:** 3 good
**Rating:** 5
**Confidence:** 4

**Summary:**

This paper introduces the idea of level sets for image classification models. Image classification models are said to be ‘under-sensitive’ when two visually distinct images produce the same output (class). The authors propose a method to compute ‘equi-cconfidence’ level sets such that two images belonging to this set produce the same output when passed through a classifier.

The Level Set Traversal (LST) algorithm takes a source image and target image (visually distinct from the source, different class) and produces an image visually similar to the target while maintaining the output prediction of the source image. This is done by iteratively updating the source image in the direction perpendicular to the gradient. The paper then goes on to show (theoretically) the behavior or level sets in some basic ML settings (linear classifier, rely neural networks etc).

Additionally, the authors also highlight the complimentary nature of adversarial examples and level sets: Adversarial examples try to change model output while keeping human predictions the same while levels sets try to keep model output the same when human predictions differ.
Experiments are done on ResNet-50 and DeiT on ImageNet. Empirical results show adversarially trained models exacerbate the problem of under sensitivity compared to vanilla models.

**Strengths:**

1. The LST algorithm is the main contribution of the paper and is a novel way to calculate equity-confidence sets given a source-target pair.
2. The paper is well written and concepts are introduced in an easy to understand fashion.
3. Significance: Understanding what causes under-sensitivity in a model is a major open problem.
4. I really like the explanation of the under-sensitivity of adversarial models using the LST algorithm. I think that is the major insight from this paper (apart from the algorithm itself of course).

Overall, I really like the idea and the execution of the paper so I'm conflicted whether to recommend acceptance  (see my comments below). I'm looking forward to hearing from the authors and seeing other reviewers' comments.

**Weaknesses:**

1. I think the main weakness of the paper is the lack of analysis around the level set found for a particular source-target pair. For example, in the adversarial example setting, adversarial examples don’t need to be w.r.t. some target, they just need to change the label with a perturbation. In this paper’s case, the level set itself depends on a target label. This isn’t necessarily an issue on it’s own, however, it is difficult to understand what showing the mere existence of a level set is supposed to show about the under-sensitivity of the model. Is there something about the level set we've found that tells us something about the under-sensitivity of the model? Other questions: How does choice of target affect the level set? Is there a ’size’ for a level set? Are level sets from source -> target and target -> source symmetric?
2. Related to presentation: I think including a figure showing images along a path from source to the final image generated by the level set algorithm would be useful to understand what exactly is within a level set. This figure could potentially replace Figure 3 which, in my opinion, does not add anything new (it shows the same thing as Figure 2 except a change in architecture)

**Questions:**

1. How are the images inside the convex hull in Fig 4 obtained? The images on the edges are calculated via linear interpolation but I am not sure about those inside the convex hull.
2. How do you distinguish ‘under-sensitivity’ from just poor training? Suppose you chose an architecture/set of hyperparameters which is bad fit for the dataset. Wouldn’t this setting also be ‘under-sensitive’? Is there any way to tell them apart?

**Limitations:**

1. (Relatively) Large computational budget required to calculate images close to the target via LST. (Mentioned by the authors)
2. Level Sets are a property of target class in addition to source class (unlike adversarial examples)

---

> ### Author Rebuttal · Authors · 2023-08-10
>
> We thank the reviewer for their insightful feedback. We are glad that they found our contributions novel, significant, and insightful. We answer the questions raised below:
>
> **Analysis and implications of level set**
> - We would like to emphasize that the level set itself is primarily a property of the source image/source class - formally, given a prediction confidence value $p \in [0,1]$, and a class $j \in \lbrace 1, \dots, N\rbrace$ the Level Set $L_f(p,j)$ for a function/model $f$, is defined as the set of all inputs that are assigned the confidence $p$ with respect to class $j$. Thus, from a theoretical standpoint, the level set itself does not depend explicitly on the choice of the target, but empirically we explore the level set using the LST algorithm which simply starts from the source image and explores in the direction of the target image. The true level set for any particular class is thus a superset of the union of the sets found by LST for all possible target images of other classes. We use random images from random target classes to empirically show that inputs very similar to the target image lie inside the level set containing the source image.
> - We agree that showing the mere existence of the level set is not very significant in itself. We however find that the level set for common vision models is remarkably expansive — large enough to contain inputs that look near-identical to target images from other classes. Further, LST helps uncover a remarkable star-like connected geometry for the level sets, wherein the linear convex interpolant paths between any source image and LST output “blindspot” image lies within the same level/super-level set. Furthermore, we show that adversarially trained models exhibit significant under-sensitivity over subsets of the input domain that lie well beyond the original threat model used in its training. The large extent of these level sets implies that we may need to think beyond adversarial training in order to solve the under-sensitivity issue.
>
> *Intermediate images along Path:* Thank you for the suggestion. We present these intermediary images found over different iterations by LST for both normal and robust ResNet-50 models in Figure-1 of the Rebuttal pdf.
>
> *Computation of images inside Convex Hull:*  In order to visualize the extent of level sets over a two-dimensional subset, we evaluate the model confidence over the triangular convex hull obtained by linearly interpolating over three reference points, namely the source image and the two target blindspot images. For example, the image at the ‘centroid’ of the triangle formed by the source image and any target image pair is the arithmetic mean of the three images. In Figure-4, the lower left vertex in each triangle is given by a source “goose” image, and the other two vertices are given by two random target blindspot images. Apart from the high-confidence assigned along the linear paths between the source and target images (represented by two sides of the triangle), we observe that the model also assigns high confidence to a sizable fraction of the interior of the triangular convex hull as well, indicating the extent of the level set.
>
> *Analyzing the 'Size' of Level Sets:*  To quantitatively characterize the extent/size of the level/super-level sets, we compute statistical measures which evaluate the model confidence on the source class over the triangular convex regions enclosed by the source image and any two target blindspot images (as shown in Figure 4 of the Main paper). In particular, we report the Average Triangle Confidence, and the fraction of images in the triangular region for which the model confidence is greater than $p_{src}−\delta$, representing different threshold confidences in Table-2 of the Main paper.
>
> *Symmetry of Level Sets:* Given a source image $x_s$ with label $y_s$ and a target image $x_t$ of class $y_t$, the level set as defined for the source image/class is mathematically distinct from that of the level set corresponding to the target image/class and are not exactly symmetric. However, by empirically running the LST algorithm with source image $x_s$ and target image $x_t$, we obtain a new input $x’$ that is perceptually similar to $x_t$ but lies within the level set of the source image $x_s$, and even the linear convex path between $x’$ and $x_s$ lie within the same level/super-level set. Symmetrically, we can rerun the LST algorithm with $x_t$ as the source while targeting $x_s$, to obtain a new image $x^”$ perceptually similar to $x_s$, though $x^”$ and the the entire linear convex path between $x^”$ and $x_t$ lie within the level set of $x_t$. Thus, though mathematically distinct, we observe similar symmetric perceptual characteristics for the two level sets empirically on common vision models.
>
> *Under-sensitivity vs Poor training:* We agree that poorly trained models will also likely display a pronounced degree of under-sensitivity. However, we can discern between these two cases by considering the network prediction of the target image itself. If the target image itself is incorrectly predicted to be in the same class as that of the source image (and say lies in its level set), then the model itself is a bad fit for these images. To control for this in our experimental evaluations, we only use target images which are correctly classified with high confidence by the network with respect to the target class. Using the LST algorithm, we show that though standard vision models may be considered a “good fit” (e.g. ResNet-50 model on ImageNet), they also simultaneously display a surprising extent of excessive under-sensitivity.
>
>
> We thank the reviewer again for their valuable comments and suggestions. We kindly ask if the reviewer would consider increasing their score to support acceptance of the paper if their concerns or questions have been addressed. We would also be glad to engage further during the author-reviewer discussion period.

---

> > ### Comment · Reviewer_LWdq · 2023-08-17
> > **Response**
> >
> > I thank the authors for their clarifying comments and updated manuscript. Thinking from a partial (Edit: practical) perspective, I'm still unsure about why discovering a star like structure (I'm picking one claim out of many about structure) is significant but I can still appreciate some theoretical insight obtained from the rest of the paper. I have accordingly raised my score to a 5.

---

> > > ### Author Response · Authors · 2023-08-18
> > > **Thank you for supporting acceptance**
> > >
> > > We are highly grateful to the reviewer for increasing the score and supporting acceptance of our paper. Regarding the significance of our methodology and results (such as star-like structure), you might find our rebuttal and official comment to reviewer QPKJ pertinent. We hope that your remaining questions regarding our work are answered.

---

> ### Comment · Area_Chair_eheq · 2023-08-16
> **Discussion?**
>
> Dear Reviewer LWdq,
>
> You provided a borderline rating before. What do you think about the authors' response? Any further questions/comments/final justifications? Some constructive discussion can really help with the reviewing process and a fair evaluation of the work.
>
> Best, Your AC

---

### Official Review · Reviewer_QPKJ · 2023-07-26

**Soundness:** 3 good
**Presentation:** 2 fair
**Contribution:** 2 fair
**Rating:** 6
**Confidence:** 4

**Summary:**

The authors aim to systematically study specific shortcomings (called blind spots) of vision models caused by their under-sensitivity.
For this purpose, they devise an algorithm that given an arbitrary pair of images (called source and target) can produce inputs that result in the same prediction output as the source image despite being "perceptually" similar to the target image (as quantified using LPIPS).
The authors study the geometry of the generated inputs and compare it under different training criteria (e.g. standard vs. adversarially robust models).


**Strengths:**

- Interesting analysis that can potentially shed light into inherent shortcomings of vision models.
- The authors provide their source code

Update after reubttal
------------------------
I would like to thank the authors for their elaborate response to the review remarks. My remarks are largely addressed in these responses, and I am hence leaning for accepting this contribution. In that case, I urge the authors to highlight the significance of their work and its utility to diagnose the sensitivity issues in adversarial training upfront. The manuscript is heavy on technical jargon that, while important, could be simplified in the main text and presented in the appendix.
As an additional suggestion, the following workshop at NeurIPS '23 would be a great fit to present some of the theoretical parts / foundations of this work https://www.neurreps.org/

**Weaknesses:**

- The contribution is rather limited. It is unclear what actionable insights we can gain using the proposed level-set analysis. There were no sufficient findings that can inform their design and training of vision models.
- The observations with respect to the "geometry of blind spots" are highly anecdotal
- The novelty is rather limited. A wide variety of methods have been devised to minimally perturb an image in order to fool the model to make a specific prediction. In the interpretability domain, various methods have meaningfully utilized minimal perturbation for the purpose of visualizing which image areas are most relevant for the input. In fact, similar perturbations are used in Integrated Gradients (Sundararajan et al. 2017), however, for a tangible application (feature attribution). See also the work by Wagner et al:
"Interpretable and Fine-Grained Visual Explanations for Convolutional Neural Networks" (CVPR '19)

Minor: There were frequent language issues. Below are the ones I noted:
- sensitivty
- dimnesional
- upto human expectations => unclear (did you mean, compared with?)
- phenonmenon
- atttacks
- near-neighbour training images => nearest-neighbour images?
- simalar



**Questions:**

How sensitive is your analysis to the parameters of the level set algorithm, besides delta (e.g. max iterations, scale factor and stepsize)?

**Limitations:**

Partially discussed

---

> ### Author Rebuttal · Authors · 2023-08-10
>
> We thank the reviewer for the detailed comments, and address the specific aspects raised below:
>
> **Contributions**
> - We present a novel Level Set Traversal (LST) algorithm that iteratively uses orthogonal components of the local gradient to identify the extent and geometry of equi-confidence sets or level sets of deep networks.
> - We emphasize that the proposed LST algorithm is applicable to general vision models such as CNNs and ViTs, unlike prior works such as Jacobsen et al. [2018a] that utilize a special class of bijective neural networks called fully Invertible RevNets, to understand and study the phenomenon of under-sensitivity of classification models.
> - Given a source image, we use LST to identify inputs (informally, "blind spots") that are very similar perceptually to arbitrary target images from other classes, while leaving the model prediction nearly unmodified.
> - We discover for the first time that such inputs ("blind spots") surprisingly are linearly connected to the original source image, that is, the network retains high confidence on the convex interpolant path as well. Thus given any source image, our work uncovers a remarkable star-like substructure within the equi-confidence level sets of common models on ImageNet and CIFAR-10, which was hitherto unknown.
> - We agree with the reviewer that at present, we cannot directly use the LST algorithm without further modifications to train vision models in order to mitigate under-sensitivity as mentioned in the Limitations section (Lines 295-300), since the images produced using a smaller budget (e.g. 10 steps) look visually distinct from the target image. We hope however that future works could tackle this problem in a computationally effective manner.
> - We draw a parallel to the field of robust defenses following the discovery of adversarial examples, which required several years to identify compute-effective techniques to mitigate the over-sensitivity of deep models, and is indeed an ongoing area of research even today. However, even the present version of the LST algorithm can help practitioners better evaluate different training methods to select an "optimal model" prior to deployment in real-world systems, even if it cannot be directly utilized during training.
>
> **Quantitative Characterisation of Under-Sensitivity**
> - To analyze the geometric structure of level sets, we demonstrate the existence of convex linear interpolant paths within the level set between the source image and the LST output quantitatively by reporting the average confidence over this linear path in Table-2, which is only marginally lower than the source image confidence itself. Since the linear interpolant path maintains high confidence for arbitrary source-target image pairs, the source image is linearly connected to the LST outputs of different target classes in a star-like substructure within the level set.
> - We then characterize the extent of the level sets by computing statistical measures which evaluate the model confidence over the triangular convex regions enclosed by the source image and any two target blindspot images. In particular, we report the Average Triangle Confidence, and the fraction of images in the triangular region for which the model confidence is greater than $p_{src}−\delta$, representing different thresholds.
> - Using these statistical metrics, we further demonstrate that adversarially trained models display a greater extent of under-sensitivity quantitatively in Table-2, as compared to normally trained models of the same architecture.
> - Backed by these quantitative measures, we firmly submit that the observations so made are not just “highly anecdotal”.
>
> **Novelty and Comparisons to Interpretability Works**
> - By utilizing orthogonal components of local gradients iteratively, LST is the first effective algorithm in exploring under-sensitivity towards arbitrary target images for vision models of generic architectures. Thus, LST helps uncover “maximal perturbations” that leave model confidence unchanged, in sharp contrast to the wide variety of adversarial attacks that minimally perturb samples to induce a significant change in the network output.
> - Moreover, in Section-1 of the Supplementary, we demonstrate that standard adversarial examples crucially are standalone insufficient to study the structure of level sets of common vision models, since model confidence along the path between a target benign sample ($x_2$) and adversarial examples (such as $x_1 + \delta_{12}$ targeted towards input $x_2$) sharply declines to a valley of near zero-confidence. This contrasts sharply with the existence of linear-connectivity observed between the source image and blind-spot inputs generated using LST.
> - To uncover important parts of a source image, the interpretability paper by Wagner et al. does engage in changing the image by computing a sparse mask formulated as a deletion/preservation game, while Integrated Gradients averages gradients along the interpolant path with respect to a “reference” all-zero or black image. In contrast however, LST finds a path with respect to a valid target image of any other class, and is meant to study the overall under-sensitivity and level set substructure in input space surrounding the source image, rather than highlighting parts of the image to interpret the network prediction at that specific input-point.
>
>
>
> We report an extensive set of quantitative and qualitative experiments for ablation analysis and robustness tests for various hyperparameters in Section-4 of the Supplementary.
>
> Typos: We thank the reviewer for pointing these out, we will amend them in future versions. We apologize for the oversight.
>
>
> We thank the reviewer again for their valuable comments and suggestions. We kindly ask if the reviewer would consider increasing their score to support acceptance of the paper if their concerns or questions have been addressed, and would be glad to engage further during the discussion period.

---

> > ### Comment · Reviewer_QPKJ · 2023-08-10
> > **Clarify the motivation**
> >
> > I appreciate the extensive responses the authors provided to all reviewers.
> > They helped me better appreciate the merits of the presented work.
> >
> > Thinking from the readers' perspective, I feel that the authors could better highlight what the motivation behind their work is, what problem it is exactly solving, why it is significant, and what follows from their results.
> >
> > The authors do introduce a variety of novel artifacts such as the triangular convex hull and the associated confidence metrics. I understand that the authors want to use these artifacts to analyze model under-sensitivity in specific OOD input regions, but it was not obvious to me what these artifacts tell us about the model's behavior, and what generalizable conclusions we can draw from them (e.g. what do star-shaped paths tell us?).
> >
> > To make my point clear, the interpolation path in Integrated Gradients was derived from two clear axioms (sensitivity and implementation invariance) instead of arbitrary definitions. Are there any parallels in the level-set solution?

---

> > > ### Author Response · Authors · 2023-08-12
> > > **Significance of Proposed Methodology**
> > >
> > > We greatly thank the reviewer for engaging in the discussion and comments. We are glad that the reviewer has appreciated our responses as well.
> > >
> > > - While the mere existence of level sets is not very significant in itself, we find that the level set for common vision models is remarkably expansive — large enough to contain inputs that look near-identical to arbitrary target images from other classes.
> > > - To put this into context, prior to this work, it was perhaps plausible to expect that if classes A and B are somewhat similar/related (such as similar-looking classes “Norfolk terrier” and “Norwich terrier” of ImageNet), then a vision model would possibly misassign high confidence for images of class B with respect to class A.
> > > - However, using the LST algorithm, we find that such high-confidence regions (say with respect to class A) extend outwards in an expansive, connected manner to include images of *arbitrary* classes, which human oracles would never state as being similar. Thus, using LST, we are able to systematically and quantitatively examine this phenomenon in common vision models for the first time.
> > > - Since the linear path from any given source image is to LST outputs for arbitrary target images retains high model confidence throughout, the level sets have a star-like substructure, where the number of “limbs” or linear protuberances of the star-like structure is extraordinarily large, plausibly as large as the number of images in all other classes.
> > > - This is significant in itself, since it indicates the hitherto unknown and unappreciated scale and extent of under-sensitivity, and moreover the relative difficulty in being able to adequately mitigate the phenomenon in practical settings. For instance, if the level set for images of class A contained sizable protuberances towards only one other class B alone, the problem could perhaps be tackled by introducing a contrastive objective during the training stage that encourages the network to better discriminate between A-B images pairs by utilizing a denser sampling of related image augmentations, likely resulting in the diminution of these specific “directed” protuberances assuming reasonable train-test generalization. But since the star-like set substructure uncovered by LST implies that such protuberances exist towards any generic image of any other class for practical networks, such simple approaches will likely be ineffective and moreover computationally infeasible from a combinatorial perspective. Thus, based on the observations uncovered with LST, we expect the problem of mitigating such a pervasive extent of under-sensitivity in common vision models to be highly non-trivial.
> > > - We utilize the triangular convex hulls to qualitatively and quantitatively analyze the size of the level sets beyond the one-dimensional interpolant paths between the source image and LST outputs alone. Thus the 1D star-like paths are contained within the triangular hulls, with the latter indicating the extent of the level sets within a two-dimensional linear subspace, for which we record relative volume at different confidence thresholds etc. in Table-2.
> > > - For instance, these triangular convex hulls help support generalizable conclusions such as that adversarial training demonstrably exacerbates under-sensitivity in a quantifiable manner, which is also readily observable qualitatively by visualizing these triangular hulls (Figure-4). Interestingly, these robust models are under-sensitive over subsets of the input domain that lie well beyond the original threat model used in its training.
> > > - Of related note is the work of Tramer et al. [2020], wherein they show that due to the misalignment of $\ell_p$ norm bounded balls and the ideal set of human-imperceptible perturbations, networks that are adversarially trained against such $\ell_p$ bounded perturbations of relatively large radius are overly-smooth *within* the same $\ell_p$ radius (more details in Section 7 of the Main paper). The triangular convex hulls and associated metrics introduced in our work indicate that under-sensitivity for robust models is extensive even when the models are trained with adversaries within a smaller radius.

---

> > > ### Author Response · Authors · 2023-08-12
> > > **Mathematical Foundations**
> > >
> > > - The theoretical underpinnings of the proposed method lie in the fact that the level set is orthogonal to the local gradient. Formally, if $\gamma(t):[0,1]\rightarrow L_g(c)$ is any differentiable curve within the level set of a differentiable real-valued function $g$, then $\frac{d}{dt}(g(\gamma(t))) = 0  = \langle \nabla g(\gamma(t)) ,\gamma'(t) \rangle $ $\forall t \in [0,1]$. Furthermore, under mild conditions it can be shown that the level set is a $(d-1)$ dimensional submanifold (see Section-2 of the main paper).
> > > - *Uniqueness:* Since the local tangent space of the level set is (d − 1) dimensional, several independent directions are orthogonal to the local gradient, and apriori do not yield a unique path like a gradient-flow. However, once we fix our target image, we can use its difference vector with respect to the source image to compute its projection onto the local tangent space to generate a *uniquely defined path within the level set*.
> > > - *Extremality:* Though this flow-based path may be non-linear, we additionally discover that the extremal point of this flow is surprisingly linearly connected with high-confidence to the source image in practical settings after we apply discretized approximations etc. Thus, if $x_s$ and $x’$ represent the source image and LST output respectively, let $\Delta x = x’ - x_s$. Then, we observe that $x_s + \alpha  \Delta x$ is assigned high confidence for all $\alpha \in [0,1]$, and $x’$ is extremal in the sense that $x_s + (1+\epsilon) \Delta x$ is rapidly assigned low-confidence by the model even for extremely small values of $\epsilon > 0$.
> > > - Thus, once the target image is fixed, outputs generated by LST enjoy two properties: (1) uniqueness and (2) extremality. While our method satisfies these properties, we do not expect it to be the unique method to do so.
> > > - In concrete settings, such as the model being (1) a linear-functional, or (2) a full-rank linear transformation, we present a detailed analysis of level sets and under-sensitivity in Section-4. We also include a discussion on the nature of level sets for ReLU networks, since they induce a piecewise linear structure over tessellated subsets of the input domain. The linear connectivity uncovered by LST consequently indicates that the overlap of different (d − 1) dimensional hyperplanes is non-trivial at a non-local scale.

---

> > > > ### Comment · Reviewer_QPKJ · 2023-08-14
> > > > **Thank you**
> > > >
> > > > I would like to thank the authors for their elaborate response, which clarified crucial aspects of the contribution. I increased my review score accordingly. As an additional suggestion, the following workshop at NeurIPS '23 would be a great fit to present some of the theoretical parts / foundations of this work https://www.neurreps.org/

---

> > > > > ### Author Response · Authors · 2023-08-15
> > > > > **Thank you**
> > > > >
> > > > > We sincerely thank the reviewer for the engaging discussion and greatly appreciate the valuable suggestions and feedback provided. We will certainly incorporate them into the final version of the paper. We thank the reviewer for the support for acceptance of the paper.
> > > > >
> > > > >
> > > > > We also thank the reviewer for the additional suggestion regarding the NeurReps Workshop, we will certainly look into it in detail.

---

### Author Rebuttal · Authors · 2023-08-10

**A note to all Reviewers**

We sincerely thank the reviewers for their time and valuable feedback on our work. We are glad that the reviewers appreciate the presentation and motivation of the proposed Level Set Traversal (LST) algorithm, its effectiveness in identifying the extent and geometry of equi-confidence level sets of deep networks, whereby subsequently uncovering the linear path connectivity and star-like substructure of these level sets. Furthermore, we are happy to note that the reviewers appreciate that the paper makes noteworthy contributions on the theoretical front towards the analysis of level set submanifolds, juxtaposed with the practical applicability of the LST algorithm to general vision models such as CNNs and ViTs (while prior approaches rely on special network architectures), thereby enabling novel analysis of the under-sensitivity and blind-spots of common models empirically on standard datasets such as ImageNet and CIFAR-10.

We greatly appreciate the valuable comments and suggestions, and we will certainly try to incorporate these in the final version of the paper.

---

### Decision · Program_Chairs · 2023-09-21

**Decision:**

Accept (spotlight)

**Comment:**

The paper presents an analysis of under-sensitivity in vision models, specifically studying blind spots and the geometry of level sets using a Level Set Traversal (LST) algorithm. The authors investigate the phenomenon of level sets in deep neural networks and propose that despite being perceptually similar, certain inputs can maintain high-confidence predictions similar to a given source image. The authors present their findings on the expansive extent of under-sensitivity in various models, contrasting normal and adversarially trained networks.

Reviewer QPKJ (giving a 6 rating) highlights the need for clearer motivation and problem statement. In their response, the authors address the reviewer's concerns regarding the motivation of their work, emphasizing the significance of their findings in uncovering the extent of under-sensitivity. They clarify the mathematical foundations of their approach, discussing uniqueness and extremity properties.

Reviewer LWdq's (giving a 5 rating) concerns regarding the significance of the discovered star-like structure and the paper's practical insights are addressed in the author's rebuttal, where they emphasize the theoretical and empirical significance of their methodology and findings. The authors explain that the level set's existence is notable due to its expansive nature and its connection to under-sensitivity, as well as its potential implications for understanding model behavior beyond adversarial training. The reviewer's remaining questions are adequately addressed in the authors' response, where they provide further explanations and clarification on the methodological and theoretical aspects of the paper. The reviewer appreciates the response and increases their score to a borderline acceptance.

Reviewer N6FN (giving a 8 rating) suggests improvements for better clarity. The authors address the need for benchmarking their approach against multiple state-of-the-art adversarial defense methods and exploring the shape of level sets across different network architectures and adversarial training methods. The authors present additional evaluation results, demonstrating compatibility with existing adversarially trained models and discussing how different architectural and training differences influence level set shapes.

Reviewer o1gw (rating 7) wants more extensive evaluation beyond the chosen 5 target images and suggest considering a metric to measure LST output confidence independently. They also suggest providing computational cost details and inquire about failure cases and best scenarios for LST. In their rebuttal, the authors address concerns about overfitting, showing that the method's effectiveness extends to various ImageNet classes. They explain the impact of hyperparameters on LST output confidence and provide additional results, including computational cost analysis. They discuss limitations related to gradient obfuscation and provide insights into best and worst-case scenarios for LST's performance. The reviewer is happy with the rebuttal and has raised their score to 7.

Review yXeg (also giving a rating 7) is advocating for paper's acceptance, and rebuttal interactions reflect a positive evaluation of the paper's contributions, clarity, and significance.

The ACs concur with the evaluations given by the reviewers and hold the view that the wide-ranging applicability of the proposed LST algorithm and the revelations it offers into neural network behavior carry significant value for the research community. This, in turn, substantiates its acceptance as a notable and impactful contribution. Therefore, the Area Chairs recommend acceptance.